# NON-DENOISING FORWARD-TIME DIFFUSIONS

## ABSTRACT

The scope of this paper is generative modeling through diffusion processes. An approach falling within this paradigm is the work of Song et al. (2021), which relies on a time-reversal argument to construct a diffusion process targeting the desired data distribution. We show that the time-reversal argument, common to all denoising diffusion probabilistic modeling proposals, is not necessary. We obtain diffusion processes targeting the desired data distribution by taking appropriate mixtures of diffusion bridges. The resulting transport is exact by construction, allows for greater flexibility in choosing the dynamics of the underlying diffusion, and can be approximated by means of a neural network via novel training objectives. We develop a unifying view of the drift adjustments corresponding to our and to time-reversal approaches and make use of this representation to inspect the inner workings of diffusion-based generative models. Finally, we leverage on scalable simulation and inference techniques common in spatial statistics to move beyond fully factorial distributions in the underlying diffusion dynamics. The methodological advances contained in this work contribute toward establishing a general framework for generative modeling based on diffusion processes.

## 1 INTRODUCTION

Denoising diffusion probabilistic modeling (DDPM) (Sohl-Dickstein et al., 2015; Ho et al., 2020; Song et al., 2021) is a recent generative modeling paradigm exhibiting strong empirical performance. Consider a dataset of $N$ samples $\mathcal{D} = \{x^{(n)}\}_{n=1}^{N}$ with empirical distribution $\mathcal{P}_{\mathcal{D}}$. The unifying key steps underlying DDPM approaches are: (i) the definition of a stochastic process with initial distribution $\mathcal{P}_{\mathcal{D}}$, whose forward-time (noising) dynamics progressively transform $\mathcal{P}_{\mathcal{D}}$ toward a simple data-independent distribution $\mathcal{P}_Z$; (ii) the derivation of the backward-time (denoising / sampling) dynamics transforming $\mathcal{P}_Z$ toward $\mathcal{P}_{\mathcal{D}}$; (iii) the approximation of the backward-time transitions by means of a neural network. Following the training step (iii), a sample whose distribution approximates $\mathcal{P}_{\mathcal{D}}$ is drawn by (iv) simulating from the approximated backward-time transitions starting with a sample from $\mathcal{P}_Z$. Both discrete-time (Ho et al., 2020) and continuous-time (Song et al., 2021) formulations of DDPM have been pursued. This work focuses on the latter case, to which we refer as diffusion time-reversal transport (DTRT). As in DTRT, dynamics are specified through diffusion processes, i.e. solutions to stochastic differential equations (SDE) with associated drift $f(\cdot)$ and diffusion $g(\cdot)$ coefficients. A number of approximations are involved in the aforementioned steps. Firstly, as the dynamics are defined on a finite time interval, a dependency from $\mathcal{P}_{\mathcal{D}}$ is retained through the noising process. Hence, starting with a sample from the data-independent distribution $\mathcal{P}_Z$ in (iv) introduces an approximation. Secondly, while the backward-time dynamics of (ii) are directly available for diffusions, they are approximated by means of a neural network in (iii). Thirdly, sampling in (iv) is achieved through a discretization on a time-grid, which introduces a discretization error. De Bortoli et al. (2021, Theorem 1) links these approximations to the total variation distance between the distribution of the generated samples from (iv) and $\mathcal{P}_{\mathcal{D}}$.

In our first methodological contribution we develop a procedure for constructing diffusion processes targeting $\mathcal{P}_{\mathcal{D}}$ without relying on time-reversal arguments. The proposed transport (coupling) between $\mathcal{P}_Z$ and $\mathcal{P}_{\mathcal{D}}$ is achieved by: (1) specifying a diffusion process $X$ on $[0, \tau]$ starting from a generic $x_0$; (2) conditioning $X$ on hitting a generic $x_\tau$ at time $\tau$, thus obtaining a diffusion bridge; (3) taking a bivariate mixture $\Pi_{0,\tau}$ of diffusion bridges over $(x_0, x_\tau)$ with marginals $\Pi_0 = \mathcal{P}_Z$ and $\Pi_\tau = \mathcal{P}_{\mathcal{D}}$, obtaining a mixture process $M$; (4) matching the marginal distribution of $M$ over $[0, \tau]$ with a diffusion process, resulting in a diffusion with initial distribution $\mathcal{P}_Z$ and terminal distribution $\mathcal{P}_{\mathcal{D}}$.

The realized diffusion bridge mixture transport (DBMT) between $\mathcal{P}_Z$ and $\mathcal{P}_\mathcal{D}$ is exact by construction. We thus sidestep the approximation common to all DDPM approaches due to the dependency from $\mathcal{P}_\mathcal{D}$ retained through the noising process. Moreover, the DBMT can be realized for almost arbitrary $\mathcal{P}_Z$, $f(\cdot)$ and $g(\cdot)$. This increased flexibility is a departure from the DTRT where $f(\cdot)$ and $g(\cdot)$ need to be chosen to obtain convergence toward a simple distribution $\mathcal{P}_Z$.

Similarly to the DTRT, achieving the DBMT requires the computation of a drift adjustment term which depends on $\mathcal{D}$. For a SDE class of interest, we develop a unified and interpretable representation of DTRT and DBMT drift adjustments as simple transformations of conditional expectations over $\mathcal{D}$. This novel result provides insights on the target mapping that we aim to approximate and on the quality of approximation achieved by the trained score models of Song et al. (2021). Having defined for the DBMT a Fisher divergence objective similarly to Song et al. (2021), we leverage on this unified representation to define two additional training objectives featuring appealing properties.

In our last methodological contribution we extend the class of SDEs that can be realistically employed in computer vision applications. Specifically, computational considerations have so far restricted the transitions of the stochastic processes employed in DDPM to be fully factorials. We view images at a given resolution as taking values over a 2D lattice which discretizes the continuous coordinate system $[0,1]^2$ representing heights and widths. Diffusion processes are viewed as spatio-temporal processes with spatial support $[0,1]^2$. Doing so, it is possible to leverage on scalable simulation and inference techniques from spatial statistics and consider more realistic diffusion transitions.

This paper is structured as follows. In Section 2 we review the DTRT of Song et al. (2021) and in Section 3 we introduce the DBMT. In order to implement the DTRT and the DBMT it is necessary to specify the underlying SDE, i.e. the coefficients $f(\cdot)$ and $g(\cdot)$. We study a class of interest in Section 4. The unified view of drift adjustments is introduced in Section 5. Section 6 develops the training objectives and Section 7 reviews the obtained results and finalizes the DBMT construction. In Section 8 we establish the connection with spatio-temporal processes. We conclude in Section 9. Appendices A to D contain the theoretical framework, assumptions, proofs, and additional material.

*Notation and conventions:* we use uppercase notation for probability distributions (measures, laws) and lowercase notation for densities; each probability distribution, and corresponding density, is uniquely identified by its associated letter not by its arguments (which are muted); for example $P(dx)$ is a distribution, $p(x)$ is its corresponding density; random elements are always uppercase (an exception is made for times, always lowercase for typographical reasons); if $P$ is the distribution of a stochastic process, we use subscript notation to refer to its finite dimensional distributions (densities with $p$), conditional or not, for some collection of times; for example $p_{t'|t}$ denotes a transition density, which is understood to be a function of four arguments $p_{t'|t}(y|x) = f(t,t',x,y)$; $\delta_x$ is the delta distribution at $x$ and $\otimes$ is used for product distributions; we refer directly to a given SDE instead of referring to the diffusion process satisfying such SDE when no ambiguity arises; we use $[a]_i$ and $[A]_{i,j}$ for vector and matrix indexing, $A^\top$ for matrix transposition.

## 2 DIFFUSION TIME-REVERSAL TRANSPORT

The starting point of Song et al. (2021) is a diffusion process $Y$ satisfying a generic $D$-dimensional time-inhomogenous SDE with initial distribution $Y_0 \sim \mathcal{P}_\mathcal{D}$

$$dY_r = f(Y_r, r)dr + g(Y_r, r)dW_r, \tag{1}$$

over noising time $r \in [0, \tau]$. Thorough this paper we denote with $Q$ the law of the diffusion solving (1) and with $q$ the corresponding densities. Thus, let $q_{r'|r}(y|x)$, $0 \le r < r' \le \tau$, be the transition density of (1), and let $q_r(y)$, $0 < r \le \tau$, be the marginal density of (1). As $Y_0 \sim \mathcal{P}_\mathcal{D}$, we have

$$q_r(y) = \frac{1}{N} \sum_{n=1}^N q_{r|0}(y|x^{(n)}). \tag{2}$$

The dynamics of (1) over the reversed, i.e. sampling, time $t = \tau - r$, $t \in [0, \tau]$, are given by (Anderson, 1982; Haussmann & Pardoux, 1986; Millet et al., 1989)

$$dX_t = [-f(X_t, r) + \nabla \cdot G(X_t, r) + G(X_t, r) \nabla_{X_t} \ln q_r(X_t)] dt + g(X_t, r)dW_t, \tag{3}$$

where $r = \tau - t$ is the remaining sampling time, $G(x, r) = g(x, r)g(x, r)^\top$ and the $D$-dimensional vector $\nabla \cdot G(x, r)$ is defined by $[\nabla \cdot G(x, r)]_i = \sum_{j=1}^D \nabla_{x_j}[G(x, r)]_{i,j}$. That is the processes $X_t$

and $Y_r = Y_{\tau-t}$ have the same distribution. Approximating the terminal distribution $Q_\tau$ of (1), i.e. the initial distribution of (3), with $\mathcal{P}_Z$, $X_0$ is sampled from $\mathcal{P}_Z$ and (3) is discretized and integrated over $t$ to produce a sample $X_\tau$ approximately distributed as $\mathcal{P}_\mathcal{D}$.

The computation of the multiplicative drift adjustment $\nabla_y \ln q_r(y)$ entering (3), i.e. the score of the marginal density (2), requires in principle $\mathcal{O}(N)$ operations. Let $s_\phi(y, r)$ be a neural network for which we would like $s_\phi(y, r) \approx \nabla_y \ln q_r(y)$. It remains to find a suitable training objective for which unbiased gradients with respect to $\phi$ can be obtained at $\mathcal{O}(1)$ cost with respect to the dataset size $N$. As $q_r(y)$ has a mixture representation, the identity of Vincent (2011) for Fisher divergences provides us with the desired objective for a fixed $r \in (0, \tau]$

$$\mathbb{L}_{\text{FD,DTRT}}(\phi, r) = \underset{Y_r \sim Q_r}{\mathbb{E}} \left[ \left\| \nabla_{Y_r} \ln q_r(Y_r) - s_\phi(Y_r, r) \right\|^2 \right]$$

$$= \underset{(Y_0, Y_r) \sim Q_{0,r}}{\mathbb{E}} \left[ \left\| \nabla_{Y_r} \ln q_{r|0}(Y_r | Y_0) - s_\phi(Y_r, r) \right\|^2 \right]. \tag{4}$$

The key point is that an unbiased, $\mathcal{O}(1)$ with respect to $N$, mini-batch Monte Carlo (MC) estimator for the expectation (4) can be trivially obtained by sampling a batch $Y_0 \sim \mathcal{P}_\mathcal{D}$, $Y_r \sim Q_{r|0}(dy_r | Y_0)$, and evaluating the average loss over the batch. In order to achieve a global approximation over the whole time interval $(0, \tau]$, Song et al. (2021) proposes uniform sampling of time $r$

$$\mathbb{L}_{\text{FD,DTRT}}(\phi) = \underset{r \sim \mathcal{U}(0,\tau], (Y_0, Y_r) \sim Q_{0,r}}{\mathbb{E}} \left[ \mathcal{R}_r \left\| \nabla_{Y_r} \ln q_{r|0}(Y_r | Y_0) - s_\phi(Y_r, r) \right\|^2 \right], \tag{5}$$

where $\mathcal{R}_r = \mathbb{E}[\|\nabla_{Y_r} \ln q_{r|0}(Y_r | Y_0)\|^2]^{-1}$ is a regularization term. A MC estimator for (5) is constructed by augmenting the MC estimator for (4) with the additional sampling step $r \sim \mathcal{U}(0, \tau]$.

## 3 DIFFUSION BRIDGE MIXTURE TRANSPORT

Our starting point is a generic $D$-dimensional time-inhomogenous SDE which, in contrast to Song et al. (2021), is directly defined on the sampling time $t \in [0, \tau]$

$$dX_t = f(X_t, t)dt + g(X_t, t)dW_t. \tag{6}$$

We reserve $P_{\cdot|0}(\cdot|x_0)$ to denote the law of the diffusion solving (6) for a given starting value $x_0$ and $p_{\cdot|0}(\cdot|x_0)$ to denote the corresponding densities.

### 3.1 DIFFUSION BRIDGES

Diffusion bridges are central to the proposed methodology, in this Section we cover their basic theory. A diffusion bridge is a diffusion process starting from a given value which is conditioned on hitting a terminal value. It is a deep result, and consequence of Doob $h$-transforms (Särkkä & Solin (2019, Chapter 7.9), Rogers & Williams (2000, Chapter IV.6.39)), that a diffusion processes pinned down on both ends is still a diffusion process. In particular the Markov property is preserved. More precisely, (6) with initial value $x_0$ conditioned on hitting a terminal value $x_\tau$ at time $\tau$ is characterized the following SDE on $[0, \tau]$ with initial value $x_0$ (Särkkä & Solin, 2019, Theorem 7.11)

$$dX_t = \left[ f(X_t, t) + G(X_t, t) \nabla_{X_t} \ln p_{\tau|t}(x_\tau | X_t) \right] dt + g(X_t, t)dW_t, \tag{7}$$

where $G(x, t) = g(x, t)g(x, t)^\top$. The multiplicative adjustment factor $\nabla_{x_t} \ln p_{\tau|t}(x_\tau | x_t)$ forces the process to hit $x_\tau$ at time $\tau$ and the diffusion process solving (7) is known as the diffusion bridge from $(x_0, 0)$ to $(x_\tau, \tau)$. As previously noted, $p_{\tau|t}(x_\tau | x_t)$ in (7) refers to the transition density of (6).

### 3.2 DIFFUSION MIXTURES

The proposed transport construction relies on a representation result for diffusion mixtures. We present here an informal version and report the precise statement, the required assumptions, and the proof in Appendix A.

**Theorem 1** (Diffusion mixture representation — informal). *Let $\{X^\lambda\}$, $\lambda \in \Lambda$ be a collection of diffusions with associated SDEs $\{dX^\lambda\}$ and marginal densities $\{\pi_t^\lambda\}$. Let $\mathcal{L}$ be a mixing distribution on $\Lambda$, $\pi_t$ be the $\mathcal{L}$-mixture of $\{\pi_t^\lambda\}$. Then there exists a diffusion process $X$ with marginal $\pi_t$. $X$ follows a SDE whose drift and diffusion coefficients are weighted averages of the corresponding coefficients in $\{dX_t^\lambda\}$, where the weights are proportional to $\{\pi_t^\lambda\}$ and to the mixing density.*

Theorem 1 is first established in Brigo (2002, Corollary 1.3) limitedly to finite mixtures and 1-dimensional diffusions. The proof of Theorem 1 in Appendix A is more direct and extends the result to the required multivariate setting. In Section 3.1 we introduced diffusion bridges mapping arbitrary initial values $x_0$ to arbitrary final values $x_\tau$. Let $\Pi_{0,\tau}$ denote a generic bivariate distribution on $\mathbb{R}^D \times \mathbb{R}^D$ with marginals $\Pi_0, \Pi_\tau$. We define the diffusion mixture $M$ as the mixture of diffusion bridges corresponding to $(X_0, X_\tau) \sim \Pi_{0,\tau}$. That is, we apply Theorem 1 to the collection of diffusion bridges (7) indexed by their initial and terminal values, $\lambda = (x_0, x_\tau)$, $\Lambda = \mathbb{R}^D \times \mathbb{R}^D$, with mixing distribution $\mathcal{L}(d\lambda) = \Pi_{0,\tau}(dx_0, dx_\tau)$. By Theorem 1 the following SDE on $[0, \tau]$ with initial distribution $\Pi_0$ has the same marginal distribution as $M$, in particular its terminal distribution is $\Pi_\tau$

$$dX_t = \mu(X_t, t)dt + g(X_t, t)dW_t,$$

$$\mu(x_t, t) = f(x_t, t) + G(x_t, t)\underbrace{\int \nabla_{x_t} \ln p_{\tau|t}(x_\tau|x_t)\frac{p_{t|0,\tau}(x_t|x_0, x_\tau)}{\pi_t(x_t)}\Pi_{0,\tau}(dx_0, dx_\tau)}_{A(x_t, t)},$$

(8)

$$\pi_t(x_t) = \int p_{t|0,\tau}(x_t|x_0, x_\tau)\Pi_{0,\tau}(dx_0, dx_\tau).$$

In (8), $A(x_t, t)$ gives the multiplicative drift adjustment factor for (6). A case of particular interest occurs when $\Pi_0$ puts all the mass on a single value $x_0$. In the following we refer to $A(x_t, t, x_0)$ in stance of $A(x_t, t)$, and to $\pi_{t|0}(x_t|x_0)$ in stance of $\pi_t(x_t)$, when it is necessary to distinguish this specific case. We also extend the scope of $\Pi$ to indicate the law of $M$. Indeed, we already denoted with $\Pi_{0,\tau}$ its initial-terminal distribution, and with $\pi_t$ its marginal density. Accordingly, $A(x_t, t) = \mathbb{E}_{X_\tau \sim \Pi(dx_\tau|x_t)}[\nabla_{x_t} \ln p_{\tau|t}(X_\tau|x_t)]$. The transport from $\mathcal{P}_Z$ to $\mathcal{P}_\mathcal{D}$ is then achieved by $\Pi_\tau = \mathcal{P}_\mathcal{D}$ and $\mathcal{P}_Z = \Pi_0$ ($\mathcal{P}_Z$ can be arbitrarily defined). As $\mathcal{P}_\mathcal{D}$ is an empirical distribution, the integral in (8) with respect to $x_\tau$ reduces to averages over $\mathcal{D}$. In summary, the diffusion $X$ solution of (8) realizes the proposed transport from $\mathcal{P}_Z$ to $\mathcal{P}_\mathcal{D}$ by matching the marginal distribution of $M$.

## 4 SDE CLASS

The starting point of the proposed transport is the unconstrained SDE (6). In this Section we define SDEs which are realized through a time-change of simpler SDEs and which are general enough to subsume the SDEs introduced in Song et al. (2021). Consider the $D$-dimensional SDEs

$$dZ_t = \Gamma^{1/2}dW_t,$$

(9)

$$dZ_t = \alpha_t Z_t dt + \Gamma^{1/2}dW_t,$$

(10)

where $\alpha_t \neq 0$ is a scalar function and $G(X_t, t) = \Gamma$ introduces an arbitrary covariance structure. (9) is the SDE of a correlated and scaled Brownian motion and (10) is the SDE of an Ornstein-Uhlenbeck process driven by a correlated and scaled Brownian motion. The transition densities of (9) and (10) are Gaussian (Appendix B). We denote both with $\widetilde{p}_{t'|t}$, informally (9) is a special case of (10) with $\alpha_t = 0$. SDE (10) in the time-homogenous case $\alpha_t = -1/2$ has stationary distribution $\mathcal{N}_D(0, \Gamma)$. We now introduce the time-change. Let $\beta_t > 0$ be a continuous function on $[0, \tau]$. Then $b_t = \int_0^t \beta_u du$ defines a monotonically (strictly) increasing function $b_t : [0, \tau] \rightarrow [0, b_\tau]$. The following SDEs on $[0, \tau]$ represent the class of dynamics for (1) and (6) on which we focus on the rest of this paper

$$dX_t = \sqrt{\beta_t}\Gamma^{1/2}dW_t,$$

(11)

$$dX_t = \alpha_t\beta_t X_t dt + \sqrt{\beta_t}\Gamma^{1/2}dW_t,$$

(12)

and $G(x, t) = \beta_t\Gamma$. The standard time-change result for diffusions (Øksendal, 2003, Theorem 8.5.1) establishes that the processes $X_t$ respectively from (11) and (12) are equivalent in law to their time-scaled counterparts $Z_{b_t}$ from (9) and (10). That is, SDEs (11) and (12) correspond to the evolution of the simpler SDEs (9) and (10) under a non-linear time wrapping where time flows with instantaneous intensity $\beta_t$. For both (11) and (12) the time-change argument yields $p_{\tau|t}(y|x) = \widetilde{p}_{b_\tau|b_t}(y|x)$ for the transition density of (6), and equivalently for the transition density $q_{\tau|t}$ of (1). We thus obtain

$$p_{\tau|t}(x_\tau|x_t) = \mathcal{N}_D(x_\tau; x_t a(t, \tau), \Gamma v(t, \tau))$$

(13)

for appropriate scalar functions $a(t, \tau), v(t, \tau)$ with $v(t, \tau) > 0$ (Appendix B). By direct computation

$$\nabla_{x_t} \ln p_{\tau|t}(x_\tau|x_t) = \Gamma^{-1}\left(\frac{x_\tau}{a(t, \tau)} - x_t\right)\frac{a^2(t, \tau)}{v(t, \tau)},$$

(14)

$$\nabla_{x_\tau} \ln p_{\tau|t}(x_\tau|x_t) = \Gamma^{-1}\left(x_t a(t,\tau) - x_\tau\right)\frac{1}{v(t,\tau)}. \tag{15}$$

From Bayes theorem and the Markov property we have

$$p_{t|0,\tau}(x_t|x_0,x_\tau) = \mathcal{N}_D\left(x_t;\ x_0\underline{a}_{\mathrm{br}}(0,t,\tau) + x_\tau\overline{a}_{\mathrm{br}}(0,t,\tau),\ \Gamma v_{\mathrm{br}}(0,t,\tau)\right), \tag{16}$$

where once again $\underline{a}_{\mathrm{br}}(0,t,\tau)$, $\overline{a}_{\mathrm{br}}(0,t,\tau)$ and $v_{\mathrm{br}}(0,t,\tau) > 0$ are scalar functions given in Appendix B. Finally, by direct computation

$$\nabla_{x_t} \ln p_{t|0,\tau}(x_t|x_0,x_\tau) = \Gamma^{-1}\frac{x_0\underline{a}_{\mathrm{br}}(0,t,\tau) + x_\tau\overline{a}_{\mathrm{br}}(0,t,\tau) - x_t}{v_{\mathrm{br}}(0,t,\tau)}. \tag{17}$$

These results provide all the analytical formulas required for the computation of the adjustment factors $A(x_t,t)$, $A(x_t,t,x_0)$ and of the training objectives used to approximate them (Section 6).

## 4.1 Interpretation of Denoising Time-Reversed SDEs

Song et al. (2021) introduces two specifications of (1), named VESDE and VPSDE, which are respectively given by

$$dY_r = \sqrt{\beta_{\mathrm{ve},r}}dW_r, \tag{18}$$

$$dY_r = -\frac{1}{2}\beta_{\mathrm{vp},r}Y_r dr + \sqrt{\beta_{\mathrm{vp},r}}dW_r. \tag{19}$$

See Appendix B for the functional form of $\beta_{\mathrm{ve},r}$ and $\beta_{\mathrm{vp},r}$. We thus recover (18) and (19) from (11) and (12) with $\Gamma = I$ and $\alpha_t = -1/2$. That is, VESDE and VPSDE correspond to a time change of the much simpler SDEs for the standard Brownian motion and for the standard Langevin SDE

$$dZ_r = dW_r,$$

$$dZ_r = -\frac{1}{2}Z_r dr + dW_r.$$

## 5 Unified View of Drift Adjustments

The linearity of SDEs (11) and (12), underlying our and Song et al. (2021) works, has the important consequence that (14) and (15) are linear in $x_t$. This in turn allow us to derive an alternative representation for the drift adjustment in (8). Indeed, substituting (14) in (8) gives (Appendix A)

$$G(x,t)A(x,t) = \beta_t\left(\frac{1}{a(t,\tau)}\underset{X_\tau\sim\Pi_{\tau|t}(dx_\tau|x)}{\mathbb{E}}[X_\tau] - x\right)\frac{a^2(t,\tau)}{v(t,\tau)}. \tag{20}$$

Similarly, for the time-reversal drift adjustment term in (3) we have (Appendix A)

$$G(x,r)\nabla_x \ln q_r(x) = \beta_r\left(a(0,r)\underset{X_\tau\sim Q_{0|r}(dx_\tau|x)}{\mathbb{E}}[X_\tau] - x\right)\frac{1}{v(0,r)}. \tag{21}$$

The relations (20) and (21) provide a unified view of the inner workings of the DTRT and of the DBMT targeting $\mathcal{P}_\mathcal{D}$. In the following we always refer to sampling time $t$. Remember that $r = \tau - t$ is the remaining sampling time. For ease of exposition we assume $\beta_t = 1$, as shown in Section 4 the term $\beta_t$ corresponds to a time-warping. The terms $a(t,\tau)$, $a(0,r)$ are "integrated scalings". They are equal to 1 for (11) and the same holds for (12) as $r \to 0$. The terms $v(t,\tau)$, $v(0,r)$ are "integrated variances". They are equal to $r$ for (11) and the same holds for (12) as $r \to 0$. We commonly refer to $E[X_\tau|x,t]$ for expectation terms in (20) and (21). Both drift adjustments (20) and (21) are thus essentially of the form $(E[X_\tau|x,t] - x)v_r^{-1}$ where the term $v_r^{-1}$ diverges as $r \to 0$.

The expectations $E[X_\tau|x,t]$ are convex linear combinations of the samples $x^{(n)}$ from $\mathcal{D}$. Explicitly, $E[X_\tau|x,t] = \sum_{n=1}^N \omega(x,t)^{(n)}x^{(n)}$, where the weights $\omega(x,t)^{(n)}$ are the probabilities, under the distributions $Q$ (time-reversal sampling process (3)) and $\Pi$ (mixture of diffusions process $M$ from Section 3.2), of reaching each state $x^{(n)}$ at terminal time $\tau$ from $x$ at time $t$. By construction, the initial weights entering expectation (20) are all equal to $1/N$ when $X$ starts from a fixed value $x_0$, and are so on average when $X_0$ is stochastic. The initial weights entering expectation (21) are

on average approximately equal to $1/N$, depending on the quality of the approximation $\mathcal{P}_Z \approx Q_\tau$. Thus, $E[X_\tau | X_0, 0]$ is an averaging of many samples $x^{(n)}$. As time progresses, changes in $X_t$ correspond to changes in $E[X_\tau | X_t, t]$ through changes in the weights $\omega(x, t)^{(n)}$. Eventually all mass concentrates on a single weight $\omega(x, t)^{(*)}$ corresponding to a dataset sample $x^{(*)}$. Ultimately, the attractor dynamics implied by $(E[X_\tau | x, t] - x)v_r^{-1}$ drive $X_t$ to $x^{(*)}$. We provide an inspection in Figure 1, where $\mathcal{D}(\text{CIFAR})$ stands for the training portion of the CIFAR10 dataset, and Euler(T) corresponds to the Euler scheme (Kloeden & Platen, 1992) applied with T discretization steps.

In the VESDE and VPSDE of Song et al. (2021) we have $G(x, r) = \beta_r I$ and reversing (21) gives

$$\mathbb{E}_{X_\tau \sim Q_{0|r}(dx_\tau | x)} [X_\tau] = \frac{v(0, r)\, \nabla_x \ln q_r(x) + x}{a(0, r)}, \tag{22}$$

where $\nabla_y \ln q_r(y)$ is the *true score*. We can thus take a *trained score model* $s_\phi(x, r) \approx \nabla_x \ln q_r(y)$, plug it in (22), and verity the extent to which $E[X_\tau | x, t]$ has been approximated, see Figure 1.

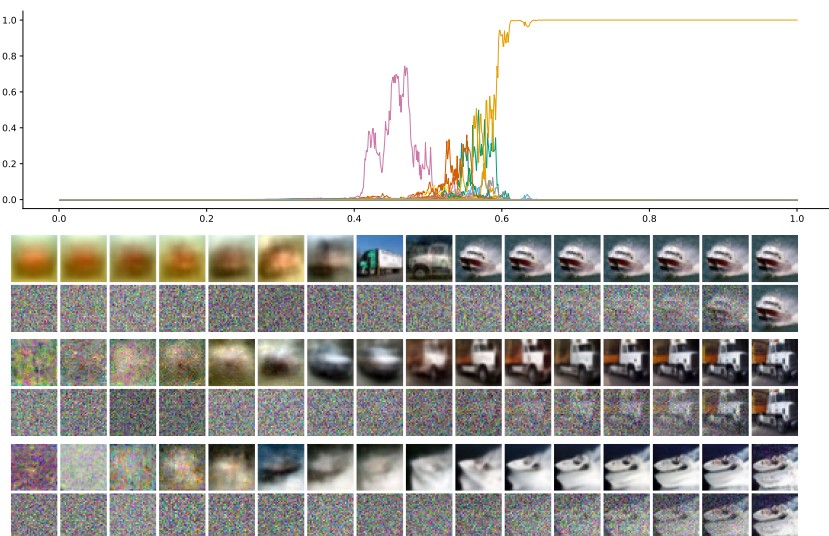

Figure 1: VPSDE model — 2nd cells' row: evolution of a trajectory of $X$ over sampling time (its terminal value $X_\tau$ is the generated sample) via the Euler(1000) discretization of (3) using the *true score* $\nabla_y \ln q_r(y)$ for $\mathcal{D}(\text{CIFAR})$; line-plot: weights' evolution $\omega(X_t, t)^{(n)}$ for all $x^{(n)}$ in $\mathcal{D}(\text{CIFAR})$ for the same $X$ (cyclical palette, many weights cannot be distinguished as they remain close to 0); 1st cells' row: $E[X_\tau | X_t, t]$ evolution for the same $X$; 3rd and 4th cells' rows: same as 1st and 2nd cells' rows for another trajectory $X$, using the *trained score model*; 5th and 6th cells' rows: same as 3rd and 4th cells' rows for another trajectory $X$, using Euler(100).

We pause for a moment to review the findings of Figure 1 (see Appendix D for additional related plots). Firstly, we can classify the dynamics of $E[X_\tau | X_t, t]$ and of the associated weights in three stages. In the 1st stage the weights do not move much. During the 2nd stage, roughly $t \in [0.4, 0.6]$, the weights' mass gets distributed over a limited number of samples. Interestingly, the weights' dynamics are not monotonic. As time progresses the weights' mass shifts between different objects from different classes. From the beginning of the 3rd stage all mass gets allocated to a single weight, the terminal image is decided well in advance of the terminal time. These dynamics are suboptimal. We would like to shorten the 1st stage, but it is associated with large values of $\beta_t$ (i.e. quick time passing) which are required to decouple $Q_\tau$ from $\mathcal{P}_\mathcal{D}$. This is an intrinsic limitation of time-reversal approaches. It is also dubious that (partially) sampling multiple objects over $t$ is beneficial for efficient generative modeling when we make use only of the terminal sample. This issue applies to trained models as well, as the 3rd row of Figure 1 shows. An interesting open question is how to obtain more suitable dynamics, where perhaps class transitions happen rarely. Secondly, $E[X_\tau | X_t, t]$ provides a denoised representation of $X_t$ across the whole 3rd stage. An alternative to the noise removal step applied to $X_\tau$ in Song et al. (2021) is to consider $E[X_\tau | X_t, t]$ as the sampling process instead. Thirdly, Figure 1 makes it clear that lowering the number of discretization steps affects generative

sampling in multiple ways. On the one hand the terminal sample $X_\tau$ is more noisy. This is not very surprising: close to $\tau$ the drift adjustment is approximately $(x^{(*)} - X_t)v_r^{-1}$, which is the drift of a Brownian bridge. Bridge sampling is notoriously problematic (Bladt et al., 2016). On the other hand larger discretization errors also significantly affect the dynamics of $E[X_\tau|X_t, t]$ resulting in less coherent samples. We remark that none of these insights could have been gained by observing $X_t$ alone, i.e. the even cells' rows of Figure 1. To conclude, (20) and (21) give an additional meaning to "denoising". Neural network approximators need to map from a noisy input $X_t$ to an adjustment toward a smoother superimposition of samples. The desire to minimize the discrepancy between the smoothness properties of $X_t$ and that of $\mathbb{E}[X_\tau|X_t, t]$ motivates the developments of Section 8.

## 6 TRANSPORTS APPROXIMATION

As in Song et al. (2021), computing the multiplicative drift adjustment $A(x_t, t)$ requires $\mathcal{O}(N)$ operations. In this Section we introduce three training objectives for which unbiased and scalable, i.e. $\mathcal{O}(1)$ with respect to $N$, MC estimators can be immediately derived.

The first training objective applies only to $A(x_t, t, x_0)$. It relies on the identity (Appendix A)

$$A(x_t, t, x_0) = \nabla_{x_t} \ln \pi_{t|0}(x_t|x_0) - \nabla_{x_t} \ln p_{t|0}(x_t|x_0). \tag{23}$$

It is advantageous to consider the right-hand side of (23) because from (8) we know that $\pi_{t|0}(x_t|x_0)$ has mixture representation. As in Song et al. (2021), we can rely on Vincent (2011) to obtain a scalable objective to train a neural network approximator $s_\phi(x_t, t) \approx \nabla_{x_t} \ln \pi_{t|0}(x_t|x_0)$, i.e.

$$\mathbb{L}_{\text{FD,DBMT}}(\phi) = \mathbb{E}_{t \sim \mathcal{U}(0,\tau), X_t \sim \Pi_{t|0}} \left[ \mathcal{J}_t \big\| \nabla_{X_t} \ln \pi_{t|0}(X_t|x_0) - s_\phi(X_t, t) \big\|^2 \right]$$

$$= \mathbb{E}_{t \sim \mathcal{U}(0,\tau), (X_t, X_\tau) \sim \Pi_{t,\tau|0}} \left[ \mathcal{J}_t \big\| \nabla_{X_t} \ln p_{t|0,\tau}(X_t|x_0, X_\tau) - s_\phi(X_t, t) \big\|^2 \right], \tag{24}$$

where $\mathcal{J}_t = \mathbb{E}[\|\nabla_{X_t} \ln p_{t|0,\tau}(X_t|x_0, X_\tau)\|^2]^{-1}$ is a regularization term.

The remaining training objectives rely on the identities (20) and (21). The goal is directly approximate the expectations of (20) and (21) which, as in Section 5, we denote with a generic $\mathbb{E}[X_\tau|x, t]$. That is, we aim to train a neural network approximator $s_\phi(x, t) \approx \mathbb{E}[X_\tau|x, t]$. As conditional expectations are mean squared error minimizers, suitable objectives for the expectation terms of (20) and (21) are

$$\mathbb{L}_{\text{CE,DBMT}}(\phi) = \mathbb{E}_{t \sim \mathcal{U}[0,\tau), (X_t, X_\tau) \sim \Pi_{t,\tau}} \left[ \big\| X_\tau - s_\phi(X_t, t) \big\|^2 \right], \tag{25}$$

$$\mathbb{L}_{\text{CE,DTRT}}(\phi) = \mathbb{E}_{r \sim \mathcal{U}(0,\tau), (Y_0, Y_r) \sim Q_{0,r}} \left[ \big\| Y_0 - s_\phi(Y_r, r) \big\|^2 \right]. \tag{26}$$

In Table 1 we summarize the operations needed to implement the plain MC estimators for the four objectives considered in this work. We reference where to find the required quantities for SDEs (11) and (12). The MC estimators for the Fisher divergence losses $\mathbb{L}_{\text{FD},*}$ involve multiplications by $\Gamma^{-1}$ (by (15) and (17)). Moreover, computing the drift adjustment at generation time requires multiplications by $\Gamma$. In Section 8 we discuss how to manage the computational burden. An appealing property of $\mathbb{L}_{\text{CE},*}$ is that computing the drift adjustment only requires the application of simple scalar functions (see (20) and (21)), and that their MC estimators only requires sampling operations. A further advantage of $\mathbb{L}_{\text{CE},*}$ is that no regularization is required. In contrast, in the absence of regularization terms, $\mathbb{L}_{\text{FD},*}$ are divergent for $t \approx \tau$ due to the term $v_r^{-1}$ (Section 5).

| $\mathbb{L}$ | Sampling ($r \sim \mathcal{U}(0,\tau], t \sim \mathcal{U}[0,\tau)$) | Evaluation |
|---|---|---|
| $\mathbb{L}_{\text{FD,DTRT}}$ | $Y_0 \sim \mathcal{P}_\mathcal{D}$, $Y_r \sim Q_{r|0}(dy_r|Y_0)$(13) | $\nabla_{Y_r} \ln q_{r|0}(Y_r|Y_0)$(15) |
| $\mathbb{L}_{\text{FD,DBMT}}$ | $X_\tau \sim \mathcal{P}_\mathcal{D}$, $(X_0 = x_0)$, $X_t \sim P_{t|0,\tau}(dx_t|x_0, X_\tau)$(16) | $\nabla_{X_t} \ln p_{t|0,\tau}(X_t|X_0, X_\tau)$(17) |
| $\mathbb{L}_{\text{CE,DTRT}}$ | $Y_0 \sim \mathcal{P}_\mathcal{D}$, $Y_r \sim Q_{r|0}(dy_r|Y_0)$(13) | |
| $\mathbb{L}_{\text{CE,DBMT}}$ | $X_\tau \sim \mathcal{P}_\mathcal{D}$, $X_0 \sim \Pi_{0|\tau}(dx_0|X_\tau)$, $X_t \sim P_{t|0,\tau}(dx_t|X_0, X_\tau)$(16) | |

Table 1: Sampling and evaluation operations required to implement the proposed MC estimators.

## 7    DBMT Overview and Numerical Experiment

In this section we finalize the DBMT construction, putting together the results of Sections 3, 4 and 6. The unconstrained SDE follows (11) or (12). It remains to choose the mixing distribution $\Pi_{0,\tau}$. The marginal $\Pi_\tau$ needs to match $\mathcal{P}_\mathcal{D}$, but there is flexibility in the choice of $\Pi_{0|\tau}$. Song et al. (2021) derived a random ordinary differential equation (RODE) matching the marginal distribution of a generative SDE, leading to faster sampling and to likelihood evaluation. RODE-matching requires $\Pi_0$ to have density. A natural implementation is given by the factorial distribution $\Pi_{0,\tau} = \mathcal{P}_Z \otimes \mathcal{P}_\mathcal{D}$ with $\mathcal{P}_Z = \mathcal{N}_D(0, \Gamma)$ and the unconstrained SDE following (12) with $\alpha_t = 1/2$ which preserves $\mathcal{P}_Z$. If instead the DBMT starts from a fixed value $x_0$, i.e. $\Pi_{0,\tau} = \delta_{x_0} \otimes \mathcal{P}_\mathcal{D}$, we can choose $x_0 = 1/N \sum_{n=1}^N x^{(n)} a(0,\tau)^{-1}$ to remove the drift adjustment at $t = 0$ (see (20)) and reduce the work required to transport $x_0$ to $\mathcal{P}_\mathcal{D}$. Finally, the use of non-factorial distributions can lead to a more efficient implementation, by linking the initial distribution to $\mathcal{P}_\mathcal{D}$.

The training steps for the simplest objective (25) of Section 6 are reported in Algorithm 1. Batch size is assumed to be 1 to ease the description. It is also assumed that $\Pi_{0,\tau}$ is factorial, otherwise the obvious modification applies to line 2 (Algorithm 2 is unaffected) where the endpoints are sampled. At line 3 a random central time and the corresponding state are sampled. The function `optimizationstep` implements a step of stochastic gradient descent update based on the loss $\mathcal{L}$. The corresponding sampling algorithm is reported in Algorithm 2 where the Euler(T) discretization is assumed in line 6. $P_{t|0,\tau}(dx_t|X_0, X_\tau)$, $a(t,\tau)$, $v(t,\tau)$, $\beta_t$ are defined in Section 4. Section 8 shows how to sample efficiently from $P_{t|0,\tau}(dx_t|X_0, X_\tau)$ and $\mathcal{N}_D(0, \Gamma)$ in computer vision applications.

---

**Algorithm 1** DBMT training ($\mathbb{L}_{\text{CE,DBMT}}$)

**Input:** $\mathcal{P}_\mathcal{D}$, $\mathcal{P}_Z$, SDE (11) or (12), NN $s_\phi(x,t)$
**Output:** trained $s_\phi(x,t)$
1: **repeat**
2:    $X_\tau \sim \mathcal{P}_\mathcal{D}$, $X_0 \sim \mathcal{P}_Z$
3:    $t \sim \mathcal{U}[0,\tau]$, $X_t \sim P_{t|0,\tau}(dx_t|X_0, X_\tau)$
4:    $\mathcal{L} \leftarrow \left\| X_\tau - s_\phi(X_t, t) \right\|^2$
5:    $\phi \leftarrow$ `optimizationstep`$(\phi, \mathcal{L})$
6: **until** convergence

**Algorithm 2** DBMT sampling ($\mathbb{L}_{\text{CE,DBMT}}$)

**Input:** $\mathcal{P}_Z$, SDE (11) or (12), trained $s_\phi(x,t)$
**Output:** Discretized path $X_{0:T}$
1: $X_0 \sim \mathcal{P}_Z$
2: **for** $s = 1, \ldots, T$ **do**
3:    $t \leftarrow (s-1)\frac{\tau}{T}$, $x \leftarrow X_{s-1}$
4:    $u_s \leftarrow \beta_t \left( \frac{1}{a(t,\tau)} s_\phi(x,t) - x \right) \frac{a^2(t,\tau)}{v(t,\tau)}$
5:    $\mathcal{E}_s \sim \mathcal{N}_D(0, \Gamma)$
6:    $X_s \leftarrow x + (f(x,t) + u_s)\frac{\tau}{T} + g(x,t)\sqrt{\frac{\tau}{T}}\mathcal{E}_s$
7: **end for**

---

We consider a toy numerical example with $\mathcal{P}_Z = \mathcal{P}_\mathcal{D} = 1/3(\delta_{-2} + \delta_0 + \delta_2)$, $D = \tau = 1$. The unconstrained SDE follows the standard Brownian motion. We consider two mixing distributions: independent mixing $\Pi_{0,1}^{\perp\!\!\!\perp}$ where $X_0$ and $X_1$ are independent and fully dependent mixing $\Pi_{0,1}^{=}$ where $X_0 = X_1$. The results are reported in Figure 2. For both couplings the correct terminal distribution $\mathcal{P}_\mathcal{D}$ is recovered, as can be seen by taking the row-wise sum of the transition matrices. The initial-terminal distribution of $X$ solving (8), which realizes the DBMT, is different from the corresponding mixing distribution $\Pi_{0,1}$ which is realized by the mixture process $M$. $\Pi_{0,1}^{\perp\!\!\!\perp}$ results in a transition matrix of equal entries $1/9$, $\Pi_{0,1}^{\perp\!\!\!\perp}$ results in a diagonal transition matrix of equal diagonal entries $1/3$.

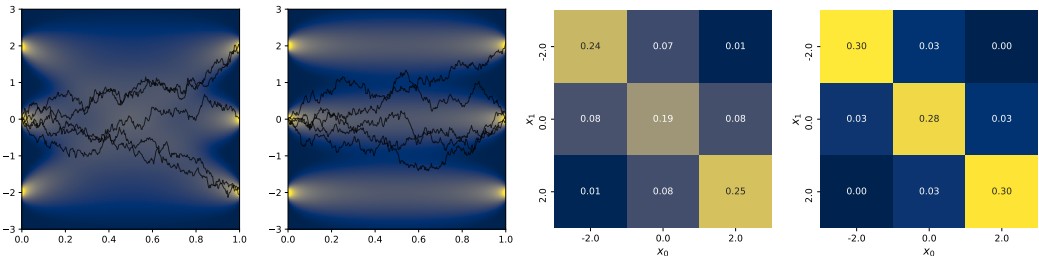

Figure 2: ($1^{\text{st}}$ ($\Pi_{0,1}^{\perp\!\!\!\perp}$), $2^{\text{nd}}$ ($\Pi_{0,1}^{=}$) plots): marginal density of the diffusion mixture $M$ in yellow, which matches the marginal density of $X$ solving (8), 5 sample paths of $X$ started at 0 in black; ($3^{\text{rd}}$ ($\Pi_{0,1}^{\perp\!\!\!\perp}$), $4^{\text{th}}$ ($\Pi_{0,1}^{=}$) plots): transition matrix of $X$ from $t = 0$ to $t = 1$ estimated from 2000 samples.

## 8 NON-DENOISING DIFFUSIONS

In computer vision applications, images of resolution $H{\times}W$ corresponds to $D = 3HW$. The use of an arbitrary covariance matrix $\Gamma$ in (11) and (12) requires its Cholesky (or equivalent) decomposition with cost $\mathcal{O}(D^3)$. As the resolution increases the computational burden gets intractable very quickly. Indeed, to the best of the authors' knowledge, all prior DDPM literature only considers independent transitions, that is $\Gamma = I$. We suggest to view SDEs (11) and (12) as corresponding to the space discretization on an $H{\times}W$ grid of a spatio-temporal process defined over the spatial domain $[0, 1]^2$. Consider the Euler discretization of (12): $X_{t+\Delta t} = X_t + \alpha_t\beta_t X_t\Delta t + \sqrt{\Delta t}\mathcal{E}_t$, where $\mathcal{E}_t \sim \mathcal{N}_D(0, \Gamma)$, the idea is to adopt a functional perspective: $X(t + \Delta t, s) = X(t, s) + \alpha_t\beta_t X(t, s)\Delta t + \sqrt{\Delta t}\mathcal{E}(t, s)$ where $s \in [0, 1]^2$ defines space coordinates. That is, both $X(t)$ and $\mathcal{E}(t)$ at each time $t$ are random processes over $[0, 1]^2$. We assume the innovations $\mathcal{E}(t)$ to be a Gaussian process (GP) for each $t$.

As the GPs $\mathcal{E}(t)$ are defined on a 2D domain we can leverage on scalable inference techniques from spatial statistics. As an example, we consider the circulant embedding method (CEM) (Wood & Chan, 1994; Dietrich & Newsam, 1997) which exploits a connection with the fast Fourier transform (FFT). See Appendix C for a cursory review of the CEM. Consider an $H{\times}W$ uniform grid $\mathcal{S}$ of size $S = HW$ discretizing $[0, 1]^2$, i.e. the support of images. For a stationary covariance function the CEM samples $\mathcal{E}(t)$ on $\mathcal{S}$ with cost $\mathcal{O}(D\ln(D))$. This is close to the $\mathcal{O}(D)$ cost of sampling from a pure white-noise process, and compares very favorably to the $\mathcal{O}(D^3)$ cost of a Cholesky decomposition. One limitation of CEM is that generated samples, while always Gaussian, might not have the correct covariances. Whether this happens, and in that case the quality of the approximation, depends on the covariance function. In Appendix C we select and fit an isotropic GP to the microscale properties of $\mathcal{D}(\text{CIFAR})$. For this estimated GP sampling is exact. Figure 3 shows samples from a pure white-noise GP, i.e. $\Gamma = I$, (1st row) and from the fitted GP using CEM (2nd row). As noted in Section 6, sampling is enough to implement the MC estimators for $\mathbb{L}_{\text{CE},*}$, but the MC estimators and drift adjustments for $\mathbb{L}_{\text{CE},*}$ involve additional matrix multiplications by $\Gamma$ and $\Gamma^{-1}$. The CEM allows to compute these at the same $\mathcal{O}(D\ln(D))$ cost if we define the GP $\mathcal{E}(t)$ on a 2D torus (Rue & Held, 2005, Chapter 2.1). This corresponds to introducing dependencies between "opposing" boundaries of $[0, 1]^2$. Figure 3 (3rd row) shows some samples, in the highlighted patch the opposing-boundaries dependency is evident. Either way, all samples from the 2nd and 3rd rows of Figure 3 match the smoothness properties of $\mathcal{D}(\text{CIFAR})$.

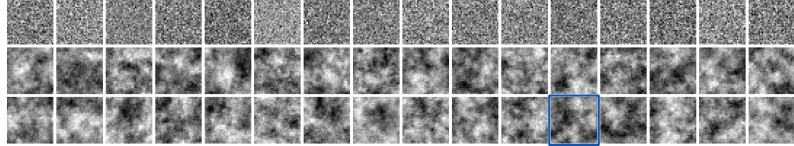

Figure 3: Spatial GP samples, see the main text for the description.

## 9 CONCLUSIONS

The DBMT construction of Section 3 is exact. The SDE class of Section 4 is tractable as it results in linear diffusion bridges. The time-space factorization of the diffusion coefficient $g(x, t) = \sqrt{\beta_t}\Gamma^{1/2}$ separates modeling concerns: $\beta_t$ corresponds to a time-wrapping, $\Gamma$ can be efficiently modeled by fitting the microscale properties of $\mathcal{P}_\mathcal{D}$. Availability of GPU-accelerated FFT implementations motivates our focus on the CEM. Alternative scalable approaches abound, from Gaussian Markov Random Fields (Rue & Tjelmeland, 2002; Rue, 2001) to Karhunen–Loève expansions (Betz et al., 2014). It remains to apply the results of this work to perform an empirical benchmarking. Section 6 develops three novel training objectives, two of which with desirable properties compared to the objective of Song et al. (2021), especially for non-factorial transitions. We remark the simplicity of the proposed DBMT approach (Algorithms 1 and 2) compared to alternatives grounded in the Schrödinger bridge problem (De Bortoli et al., 2021; Wang et al., 2021; Vargas et al., 2021). The understanding of the target mappings ((20) and (21)) can guide the development of neural networks more closely matching the target structure compared to the U-Net default choice. `https://github.com/?` links to the code accompanying this paper which is made available under the `MIT` license.

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

## A   THEORETICAL FRAMEWORK

### A.1   ASSUMPTIONS

**Assumption 1** (SDE solution). *A given $D$-dimensional SDE($f, g$) with associated initial distribution $\mathcal{V}_0$ and integration interval $[0, \tau]$ admits a unique strong solution on $[0, \tau]$.*

Assumption 1 can be checked through the application of the standard existence and uniqueness theorems for SDE solutions. Of particular relevance to our setting is the formulation of Krylov (1995, Chapter 5, Theorem 1) that limits the monotonic requirement to, informally speaking, drifts that pull the process toward infinities.

**Assumption 2** (SDE density). *A given $D$-dimensional SDE($f, g$) with associated initial distribution $\mathcal{V}_0$ and integration interval $[0, \tau]$ admits a marginal / transition density on $(0, \tau)$ with respect to the $D$-dimensional Lebesgue measure that uniquely satisfies the Fokker-Plank / Kolmogorov-forward partial differential equation (PDE).*

We refer to Särkkä & Solin (2019, Chapter 5) and to Karatzas & Shreve (1996, Chapter 5.7) for connections between SDEs and PDEs.

All theoretical results of this work rely on simple algebraic manipulations and re-arrangements of quantities of interest. The main complication stems from the need to justify differentiation and integration exchanges, i.e. exchange of limits.

**Assumption 3** (exchange of limits). *We assume that limits exchanges are justified in the steps marked with $(\star)$ and $(\star\star)$.*

Similarly, various steps in the derivations involve considering fractional quantities with densities appearing in the denominators.

**Assumption 4** (positivity). *For a given stochastic process, all finite-dimensional densities, conditional or not, are strictly positive.*

Assumption 4 is easy to verify. We resorted to the practical but somewhat unsatisfactory formulation of Assumption 3 because in full generality it is complicated to give easy to check conditions. We just note that when $\Pi_T = \mathcal{P}_\mathcal{D}$, the limit exchange marked with $(\star)$ is always justified. So are the limits exchanges marked with $(\star\star)$ when in addition $\Pi_0$ puts all the mass to a fixed initial value, or (by direct verification) when $\Pi_0$ is Gaussian for the SDE class of Section 4. Thorough this paper, both in the main text and in the proofs that follow, it is supposed that Assumptions 1, 2 and 4 are satisfied by SDEs (1), (6) and (7). This is the case for the SDE class of Section 4, i.e. (11) and (12), for any $\Pi_0$ with finite variance.

*Remark:* For ease of exposition it is assumed thorough this paper that all diffusions take values in the state space $\mathbb{R}^D$. There is no impediment in extending the presented results to the case of diffusions taking values in a subset $\mathcal{X} \subset \mathbb{R}^D$. The obvious changes to Assumptions 1 to 4 apply, the proofs carry over without substantial modifications. This extension could be of practical interest as images are often represented as floating point values in $[0, 1]$.

### A.2   STATEMENT AND PROOF OF DIFFUSION MIXTURE REPRESENTATION THEOREM

**Theorem 2** (Diffusion mixture representation). *Consider the family of $D$-dimensional SDEs on $t \in [0, \tau]$ indexed by $\lambda \in \Lambda$*

$$dX_t^\lambda = \mu^\lambda(X_t^\lambda, t)dt + \sigma^\lambda(X_t^\lambda, t)dW_t^\lambda,$$
$$X_0^\lambda \sim \mathcal{V}_0^\lambda, \tag{27}$$

*where the initial distributions $\mathcal{V}_0^\lambda$ and the BMs $W_t^\lambda$ are all independent. Let $\nu_t^\lambda, t \in (0, \tau)$ denote the marginal density of $X_t^\lambda$. For a generic mixing distribution $\mathcal{L}$ on $\Lambda$, define the mixture marginal density $\nu_t$ for $t \in (0, \tau)$ and the mixture initial distribution $\mathcal{V}_0$ by*

$$\nu_t(x) = \int_\Lambda \nu_t^\lambda(x)\mathcal{L}(d\lambda), \quad \mathcal{V}_0(dx) = \int_\Lambda \mathcal{V}_0^\lambda(dx)\mathcal{L}(d\lambda). \tag{28}$$

*Consider the $D$-dimensional SDE on $t \in [0, \tau]$ defined by*

$$\mu(x, t) = \frac{\int_\Lambda \mu^\lambda(x, t) \nu_t^\lambda(x) \mathcal{L}(d\lambda)}{\nu_t(x)},$$

$$\sigma(x, t) = \frac{\int_\Lambda \sigma^\lambda(x, t) \nu_t^\lambda(x) \mathcal{L}(d\lambda)}{\nu_t(x)}, \tag{29}$$

$$dX_t = \mu(X_t, t)dt + \sigma(X_t, t)dW_t,$$

$$Y_0 \sim \mathcal{V}_0.$$

*It is assumed that all diffusion processes $X^\lambda$ and the diffusion process $X$ solving (29) satisfy the regularity assumptions Assumptions 1, 2 and 4 and that Assumption 3 holds. Then the marginal distribution of the diffusion $X$ is $\nu_t$.*

*Proof of Theorem 2.* We start by establishing that the law of $X$ is indeed given by the solution of (29). In this proof we make use of the following notation: for $f$ scalar-valued $(f)_t = \frac{d}{dt} f$, for $a$ vector-valued $(a)_x = \sum_{i=1}^D \frac{d}{dx_i} a$, for $A$ matrix-valued $(A)_{xx} = \sum_{i,j=1}^D \frac{d^2}{dx_i dx_j} A$. This notation allows for a compact representation of PDEs reminiscent of the 1-dimensional setting. Then, for $0 < t < \tau$ we have that

$$
\begin{aligned}
(\nu(x, t))_t &= \left( \int_\Lambda \nu^\lambda(x, t) \mathcal{L}(d\lambda) \right)_t \\
&= \int_\Lambda \left( \nu^\lambda(x, t) \right)_t \mathcal{L}(d\lambda) \quad\quad (\star\star) \\
&= \int_\Lambda \left( \mu^\lambda(x, t) \nu^\lambda(x, t) \right)_x + \frac{1}{2} \left( \sigma^\lambda(x, t) \nu^\lambda(x, t) \right)_{xx} \mathcal{L}(d\lambda) \\
&= \int_\Lambda \left( \frac{\mu^\lambda(x, t) \nu^\lambda(x, t)}{\nu(x, t)} \nu(x, t) \right)_x + \frac{1}{2} \left( \frac{\sigma^\lambda(x, t) \nu^\lambda(x, t)}{\nu(x, t)} \nu(x, t) \right)_{xx} \mathcal{L}(d\lambda) \\
&= \left( \int_\Lambda \frac{\mu^\lambda(x, t) \nu^\lambda(x, t)}{\nu(x, t)} \mathcal{L}(d\lambda) \nu(x, t) \right)_x + \frac{1}{2} \left( \int_\Lambda \frac{\sigma^\lambda(x, t) \nu^\lambda(x, t)}{\nu(x, t)} \mathcal{L}(d\lambda) \nu(x, t) \right)_{xx}. \quad (\star\star)
\end{aligned}
$$

The second line is an exchange of limits, the third line is the application of the Fokker-Plank PDEs for the collection of processes $X^\lambda$, the fourth line is a rewriting in terms of $\nu(y, t)$, the last line is another exchange of limits. The result follows by noticing that the last line gives the Fokker-Plank representation of (29). $\square$

## A.3 DRIFT ADJUSTMENTS

Limitedly to this section, we lighten the notation by removing subscripts from probability measures and densities. The missing time points can be inferred without ambiguity from the variables.

### A.3.1 DRIFT ADJUSTMENT IDENTITIES FOR CONSTANT INITIAL VALUE

First identity:

$$
\begin{aligned}
&\nabla_{x_t} \ln \int \frac{p(x_\tau | x_t)}{p(x_\tau | x_0)} \Pi(dx_\tau) \\
&= \int \frac{\nabla_{x_t} p(x_\tau | x_t)}{p(x_\tau | x_0)} p(x_t | x_0) \Pi(dx_\tau) \Big/ \int \frac{p(x_\tau | x_t)}{p(x_\tau | x_0)} p(x_t | x_0) \Pi(dx_\tau) \quad (\star) \\
&= \int \nabla_{x_t} \ln p(x_\tau | x_t) \frac{p(x_\tau, x_t | x_0)}{p(x_\tau | x_0)} \Pi(dx_\tau) \Big/ \int \frac{p(x_\tau, x_t | x_0)}{p(x_\tau | x_0)} \Pi(dx_\tau) \\
&= \int \nabla_{x_t} \ln p(x_\tau | x_t) p(x_t | x_0, x_\tau) \Pi(dx_\tau) \Big/ \pi(x_t | x_0) \\
&= A(x_t, t, x_0).
\end{aligned}
$$

Second identity:

$$\nabla_{x_t} \ln \int \frac{p(x_\tau|x_t)}{p(x_\tau|x_0)} \Pi(dx_\tau)$$

$$= \nabla_{x_t} \ln \int \frac{p(x_\tau|x_t)}{p(x_\tau|x_0)} p(x_t|x_0) \Pi(dx_\tau) - \nabla_{x_t} \ln p(x_t|x_0)$$

$$= \nabla_{x_t} \ln \pi(x_t|x_0) - \nabla_{x_t} \ln p(x_t|x_0).$$

### A.3.2 DRIFT ADJUSTMENTS AS EXPECTATIONS

To establish (20) notice that from (14) we have

$$G(x_t, t)A(x_t, t)$$

$$= \beta_t \Gamma \Gamma^{-1} \int \left( \frac{x_\tau}{a(t,\tau)} - x_t \right) \frac{a^2(t,\tau)}{v(t,\tau)} \frac{p(x_t|x_0, x_\tau)}{\pi(x_t)} \Pi_{0,\tau}(dx_0, dx_\tau)$$

$$= \beta_t \left( \frac{1}{a(t,\tau)} \int x_\tau \frac{\pi(x_t|x_0, x_\tau)}{\pi(x_t)} \Pi_{0,\tau}(dx_0, dx_\tau) - x_t \right) \frac{a^2(t,\tau)}{v(t,\tau)}$$

$$= \beta_t \left( \frac{1}{a(t,\tau)} \mathop{\mathbb{E}}_{X_\tau \sim \Pi(dx_\tau|x_t)} [X_\tau] - x_t \right) \frac{a^2(t,\tau)}{v(t,\tau)}.$$

To establish (21) note that from (15) we have

$$G(y_r, r) \nabla_{y_r} \ln q(y_r) = \Gamma \int \nabla_{y_r} \ln q(y_r|y_0) \frac{q(y_r|y_0)}{q(y_r)} \mathcal{P}_\mathcal{D}(dy_0)$$

$$= \beta_r \Gamma \Gamma^{-1} \int \left( \frac{a(0,r)y_0 - y_r}{v(0,r)} \right) \frac{q(y_r|y_0)}{q(y_r)} \mathcal{P}_\mathcal{D}(dy_0)$$

$$= \beta_r \left( a(0,r) \int y_0 \frac{q(y_r|y_0)}{q(y_r)} \mathcal{P}_\mathcal{D}(dy_0) - y_r \right) \frac{1}{v(0,r)}$$

$$= \beta_r \left( a(0,r) \mathop{\mathbb{E}}_{Y_0 \sim Q(dx_0|y_r)} [Y_0] - y_r \right) \frac{1}{v(0,r)}.$$

## B SDEs CLASS FORMULAS

The transition densities of (9) and (10) are given respectively by

$$\widetilde{p}_{\mathrm{bm},\tau|t}(z_\tau|z_t) = \mathcal{N}_D \left( z_\tau;\, z_t,\, \Gamma(\tau - t) \right),$$

$$\widetilde{p}_{\mathrm{ou},\tau|t}(z_\tau|z_t) = \mathcal{N}_D \left( z_\tau;\, z_t e^{\overline{\alpha}_{t:\tau}(\tau - t)},\, \Gamma \left( \frac{1}{2\alpha_t} e^{2\overline{\alpha}_{t:\tau}(\tau - t)} - \frac{1}{2\alpha_\tau} \right) \right).$$

Here we used the notation $\overline{f}_{t:\tau} = \frac{1}{\tau - t} \int_t^\tau f_u du$, i.e. $\overline{f}_{t:\tau}$ is the average value of a function $f_u$ on the interval $[t, \tau]$. The time-homogenous case of (10), where $\overline{\alpha}_{t:\tau} = \alpha$, is thus immediately recovered.

The scalar functions $a_{\mathrm{bm}}(t,\tau)$, $a_{\mathrm{ou}}(t,\tau)$, $v_{\mathrm{bm}}(t,\tau)$ and $v_{\mathrm{ou}}(t,\tau)$ are given by

$$a_{\mathrm{bm}}(t,\tau) = 1, \qquad\qquad\qquad v_{\mathrm{bm}}(t,\tau) = b_\tau - b_t,$$

$$a_{\mathrm{ou}}(t,\tau) = e^{\overline{\alpha}_{b_t:b_\tau}(b_\tau - b_t)}, \qquad\qquad v_{\mathrm{ou}}(t,\tau) = \frac{1}{2\alpha_{b_t}} e^{2\overline{\alpha}_{b_t:b_\tau}(b_\tau - b_t)} - \frac{1}{2\alpha_{b_\tau}}.$$

The scalar functions $v_{\mathrm{br}}(0,t,\tau)$, $\underline{a}_{\mathrm{br}}(0,t,\tau)$ and $\overline{a}_{\mathrm{br}}(0,t,\tau)$ are given by

$$v_{\mathrm{br}}(0,t,\tau) = \frac{v(0,t)v(t,\tau)}{v(0,t)a^2(t,\tau) + v(t,\tau)},$$

$$\underline{a}_{\mathrm{br}}(0,t,\tau) = \frac{v(t,\tau)a(0,t)}{v(0,t)a^2(t,\tau) + v(t,\tau)},$$

$$\overline{a}_{\mathrm{br}}(0,t,\tau) = \frac{v(0,t)a(t,\tau)}{v(0,t)a^2(t,\tau) + v(t,\tau)}.$$

The scalar functions $\beta_{\text{ve},r}$ and $\beta_{\text{vp},r}$ are given by

$$\beta_{\text{ve},r} = \sigma_{\min}^2 \left( \frac{\sigma_{\max}}{\sigma_{\min}} \right)^{2r} 2 \log \frac{\sigma_{\max}}{\sigma_{\min}}, \quad \beta_{\text{vp},r} = \left( \bar{\beta}_{\min} + r \left( \bar{\beta}_{\max} - \bar{\beta}_{\min} \right) \right).$$

The constants $\sigma_{\min}, \sigma_{\max}, \bar{\beta}_{\min}, \bar{\beta}_{\max}$ depend in part on the dataset considered, but are consistently chosen to have $\beta_{\text{ve},r}, \beta_{\text{vp},r}$ small for $r \approx 0$ and large for $r \approx \tau$.

The approximating distributions $\mathcal{P}_Z$ in Song et al. (2021) are $\mathcal{P}_Z^{\text{ve}} = \mathcal{N}_D(0, I\sigma_{\max}^2)$ for VESDE, $\mathcal{P}_Z^{\text{vp}} = \mathcal{N}_D(0, I)$ for VPSDE.

## C  ADDITIONAL MATERIAL

### C.1  CLOSELY RELATED WORK

A work closely related to the present paper is that of Wang et al. (2021) as it similarly avoids the time-reversal construction. Wang et al. (2021) construct a 2-stages diffusion process from a constant initial value $x_0$ to $\mathcal{P}_D$ by relying on the theory of Schrödinger bridges. The most notable differences with respect to the DBMT transport are: (i) the dynamics considered in Wang et al. (2021) are less general, in our notation they correspond to $f(\cdot) = 0, g(\cdot) = \sigma I$ for a fixed scalar $\sigma$; (ii) the transport proposed in Wang et al. (2021) necessarily starts from $x_0$, the general result of (8) allows for (almost) arbitrary initial distributions and initial-terminal dependencies. For an initial $x_0$, i.e. for case of $A(x_t, t_x 0)$ in Section 3.2 and for the more limited dynamics considered in Wang et al. (2021), the achieved transport is the same. In this sense the DBMT generalizes the first stage diffusion of Wang et al. (2021). It is interesting to note that in the case of a constant $x_0$ the DBMT can also be obtained by an application of Doob $h$-transforms as we show in the following section.

We now review two additional works grounded in the Schrödinger bridge problem: De Bortoli et al. (2021); Vargas et al. (2021). Both works rely on the Iterative Proportional Fitting (IPF) procedure to solve the (dynamic) Schrödinger bridge problem. Both works leverage on time-reversal results to carry out the alternated Schrödinger half-bridge IPF iterations. The main difference between the two works is that De Bortoli et al. (2021) estimates the optimal SDE drifts via neural network approximations and score-matching, while Vargas et al. (2021) relies on Gaussian Processes and maximum likelihood fitting. The work of De Bortoli et al. (2021) can be seen as an extension of Song et al. (2021), and similarly to our work allows the use of shorter time intervals. Compared to our proposal, it solves a harder problem but also presents additional difficulties. Training is more involved as all the neural network approximations, one for each IPF iterate, need to converge. Moreover, there is limited guidance on how to optimally choose the number of integration steps over the number of IPF iterates.

### C.2  CONNECTION WITH DOOB h-TRANSFORMS

The previously established identity

$$A(x_t, t, x_0) = \nabla_{x_t} \ln \int \frac{p_{\tau|t}(x_\tau | x_t)}{p_{\tau|0}(x_\tau | x_0)} \Pi_\tau(dx_\tau),$$

shows that the drift adjustment can be equivalently expressed as

$$\mu(x_t, t) = f(x_t, t) + G(x_t, t) \nabla_{x_t} h(x_t, t), \quad h(x_t, t) = \ln \int \frac{p_{\tau|t}(x_\tau | x_t)}{p_{\tau|0}(x_\tau | x_0)} \Pi_\tau(dx_\tau),$$

as $x_0$ is a constant. It can be verified that the $h$ function satisfies the required space-time regularity property (Särkkä & Solin, 2019, Eq. (7.73)). As such, it is a genuine Doob $h$-transform. That $p_{t'|t}^h(x_{t'} | x_t) = p_{t'|t}(x_{t'} | x_t) h(x_{t'}, t') / h(x_t, t)$ is the transition density of the DBMT transport from $\delta_{x_0}$ to $\Pi_\tau$ follows by direct computation.

### C.3  GP MODELLING ON CIFAR10

For simplicity, we assume a factorial distribution over the channels and an isotopic stationary covariance function. We rely on the semivariogram approach (Cressie, 1993) to compare how

different covariance functions fit $\mathcal{D}(\text{CIFAR})$. A semivariogram is a measure of dependency across space. In the case of an isotropic stationary covariance it simplifies to a scalar function of the Euclidean distance between points: $\gamma(\|\Delta s\|) = \mathbb{E}[(\Delta x_s)^2]/2$ with $\Delta x_s = x_{s+\Delta s} - x_s$. The rate of decrease of $\gamma(\|\Delta s\|)$ toward 0 as $\|\Delta s\| \to 0$ gives a measure of the infinitesimal spatial dependency, i.e. the smoothness of the spatial process. Semivariograms corresponding to different covariance functions are here fitted to their empirical counterparts via a weighted minimum-least-squares procedure (Cressie, 1993). Figure 4 illustrates the exponential and RBF semivariogram fits for two images of $\mathcal{D}(\text{CIFAR})$. The exponential covariance, which corresponds to rougher paths, provides a much better fit to the shown samples. This result is consistent across $\mathcal{D}(\text{CIFAR})$. We remark that $\Gamma = I$ corresponds to a pure white-noise process with a perfectly flat semivariogram which would clearly result in a very poor fit to the empirical semivariograms shown in Figure 4. Based on these findings, we model the innovations of each image channel as a GP with exponential covariance function with length-scale $\theta = 0.205$, the median estimated value (Figure 4 (right)). We match the marginal variance to that of $\mathcal{D}(\text{CIFAR})$, $\sigma^2 = 0.063$.

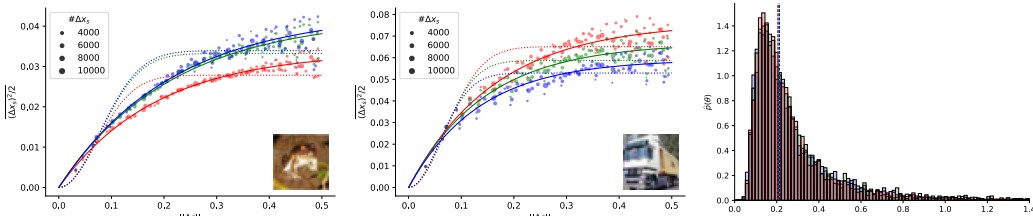

Figure 4: Empirical semivariograms (dots) and fitted exponential (solid lines) and RBF (doted lines) semivariograms for the 1$^{\text{st}}$ (left) and 2$^{\text{nd}}$ (center) image of $\mathcal{D}(\text{CIFAR})$; histograms (bins) and medians (lines) of the distributions of the length-scale parameters in the exponential variogram model over $\mathcal{D}(\text{CIFAR})$ (right); colors represent the RGB channels.

### C.4 CIRCULANT EMBEDDING METHOD

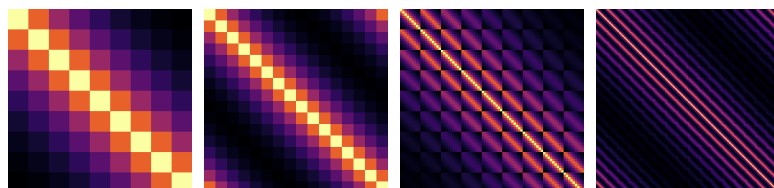

Figure 5: Circulant embedding covariance matrices, see the appendix's main text for the description.

We start by providing a cursory explanation leading to efficient sampling in the 1D case, before giving the intuition behind the extension to the 2D case. We refer to Wood & Chan (1994); Dietrich & Newsam (1997) for a complete explanation and to Rue & Held (2005) for the results underlying efficient density (likelihood) computation. Let $[0, 1]$ be the spatial domain of interest. Let $\mathcal{S} = \{s_i\}_{i=1}^M$ be a uniform grid (regular lattice) discretizing $[0, 1]$, where the points $s_i$ are assumed to be ordered. The first key observation is that for a stationary covariance function $\rho(\cdot, \cdot)$ the covariance matrix $C$ with entries $C_{i,j} = \rho(s_j, s_j)$ is symmetric and Toeplitz, i.e. with constant-diagonals. See Figure 5 (leftmost) for an example where $M = 9$. A property of symmetric Toeplitz matrices is that they can always be embedded in larger symmetric circulant matrices. A circulant matrix of size $M' \times M'$ is defined by the property that all its rows (and columns) are obtained by cycling through the same $M'$-dimensional vector. Circulant matrices correspond to covariance matrices of GPs defined on a (here 1D) torus (Rue & Held, 2005, Chapter 2.1). The circulant embedding matrix just introduced corresponds to an artificial enlargement of the spatial domain $[0, 1]$ to a larger interval leading to a torus. See Figure 5 (2$^{\text{nd}}$ from left) for a circulant embedding of $C$. The second key observation is that a circulant matrix is diagonalized by the 1D FFT matrix. Having obtained the eigenvalues of $C$, efficient sampling on the enlarged domain is achieved by the 1D FFT applied to

complex standard random numbers multiplied by the (square root of the) eigenvalues. The real and imaginary part of the generated samples are independent. The main issue with the CEM is that the circulant matrix embedding might fail to be positive definite. The issue can be avoided by considering progressively larger embeddings, see the theoretical and empirical findings of Dietrich & Newsam (1997). Otherwise, a level of approximation can be accepted by modifying the covariance function or by truncating the eigenvalues to be positive.

The development of the 2D CEM follows very similar steps. The domain of interest is now $[0, 1]^2$, the uniform grid is $\mathcal{S} = \{s_{i,j}\}_{i,j=1}^M$ and the points $s_{i,j}$ are assumed to be lexicographically ordered. The stationarity of the covariance functions results in a symmetric block-Toeplitz covariance matrix, as shown in Figure 5 (3rd from left). Again, symmetric block-Toeplitz matrices can be embedded in symmetric block-circulant matrices, as exemplified by Figure 5 (rightmost, zooming might be required to see the block structure). Block-circulant matrices can be shown to be diagonalized by the 2D FFT matrix, and efficient sampling follows from similar steps to the ones seen in the 1D case.

# D  ADDITIONAL FIGURES

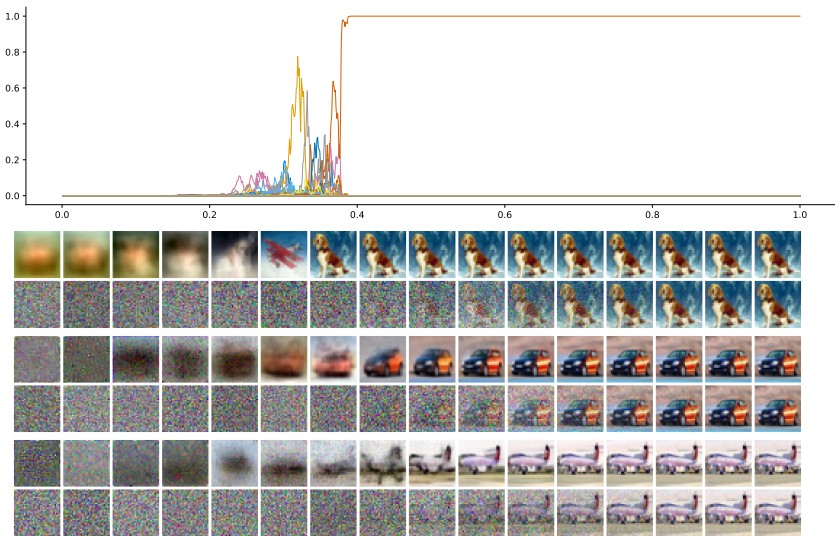

Figure 6: Same as Figure 1 for VESDE model.

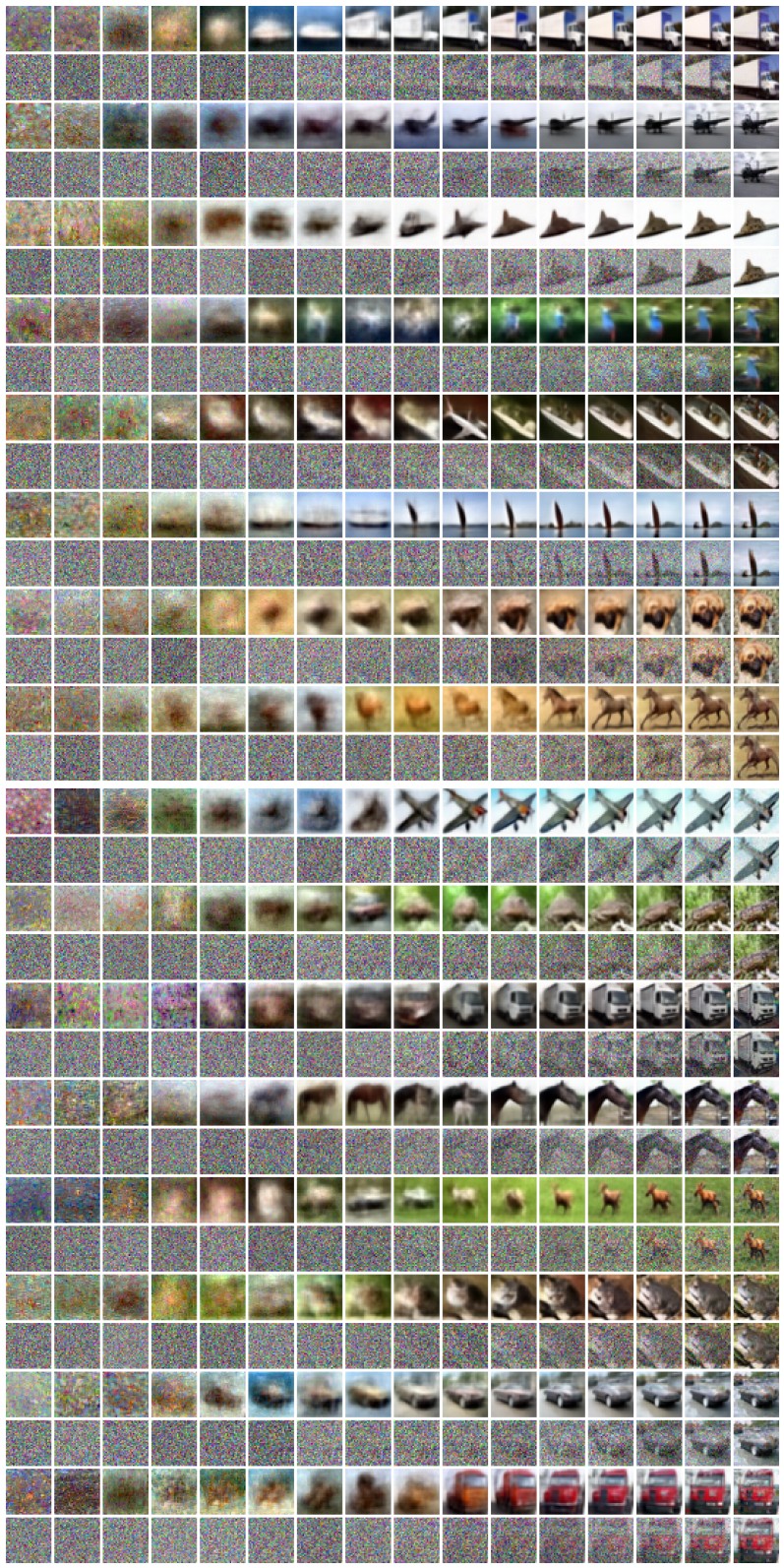

Figure 7: Additional samples from the trained VPSDE model of Song et al. (2021), $E[X_\tau | X_t, t]$ and $X_t$ (interleaved rows) over sampling time $t$, Euler(1000) (top 16 rows) and Euler(100) (bottom 16 rows).

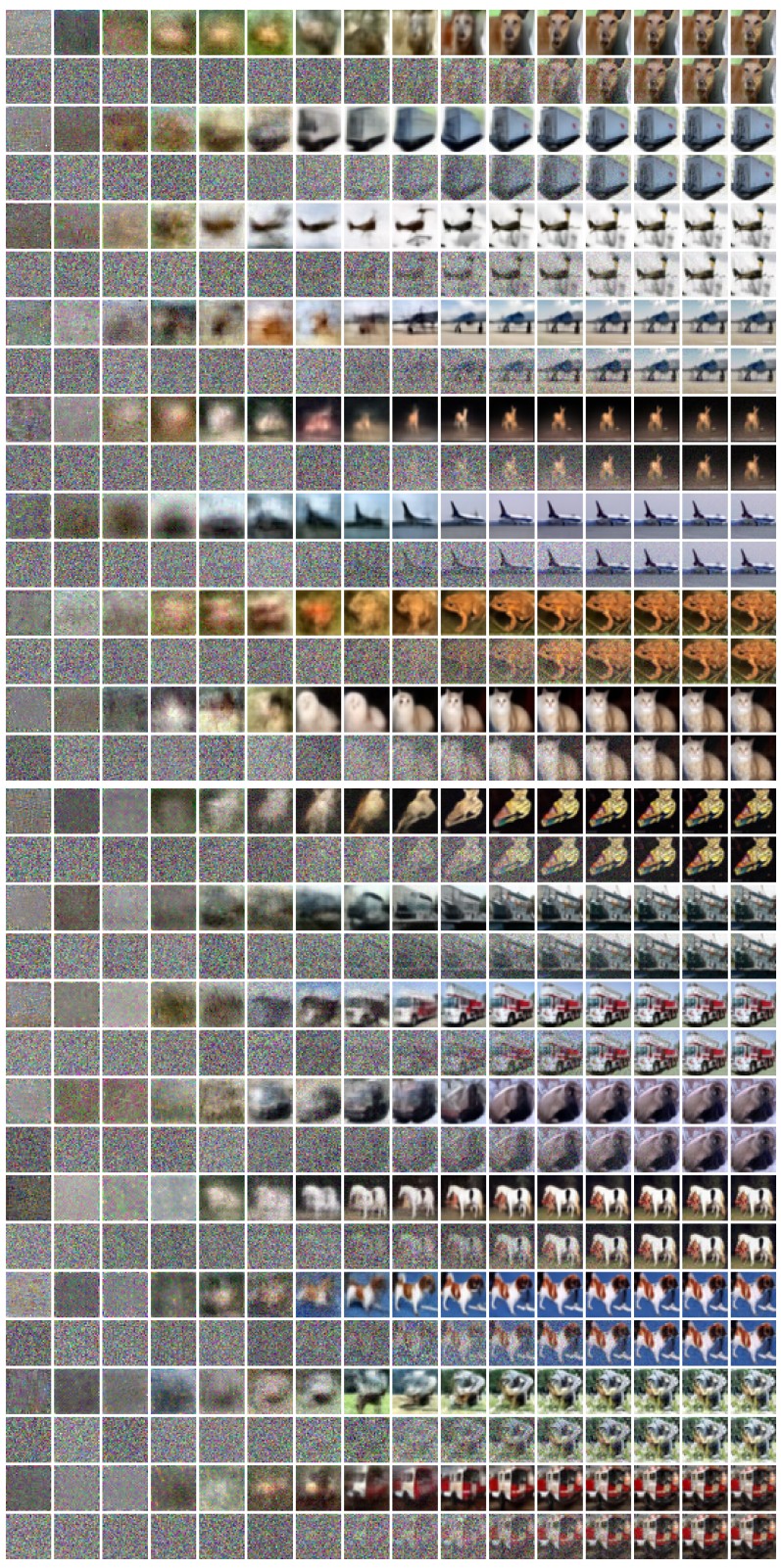

Figure 8: Additional samples from the trained VESDE model of Song et al. (2021), $E[X_\tau|X_t, t]$ and $X_t$ (interleaved rows) over sampling time $t$, Euler(1000) (top 16 rows) and Euler(100) (bottom 16 rows).

