# OpenReview forum: "Non-Denoising Forward-Time Diffusions"
_ICLR.cc/2022/Conference — ICLR 2022 Submitted_

### Official Review · Reviewer_e93q · 2021-10-30

**Correctness:** 3
**Technical Novelty And Significance:** 2
**Empirical Novelty And Significance:** 2
**Recommendation:** 5
**Confidence:** 2

**Main Review:**

I think this paper is written in an obscure way for the non-specialists of this field (despite I have some limited background in SDEs). In many places, the motivations and constructions are not clear and not written precise enough for the ICLR readers.

My comments are as follows:

1) The authors need to derive Eq. (7) within the paper for completeness. This result is deferred to several other theorems but it is hard to check whether assumptions made in these works to prove these theorems hold in this setting. What kind  of assumptions underlie the original result? Do they hold?

2) I wonder why authors move from Eq. (7) to mixtures, as it is not made clear. Why is it not enough to use Eq. (7) for the generative modelling?

3) In Eq. (8) where authors introduce the SDE, $g(X_t, t)$ is not specified.

4) I found Section 4 quite confusing. The authors seem to define in this section an SDE class which are realised through a time-change of simpler SDEs (as they put it), I really couldn’t see how this relates to the rest of the paper. The authors should clarify what they mean by “the following SDEs represent the class of dynamics for (1) and (6)”. If this means that these SDEs, via a change of time argument, can represent (6), this notion has to be made clearer.

Also, these results need to be proved clearly instead of referring to the Oksendal’s book as done in the paper.

**Summary Of The Paper:**

This paper proposes a diffusion bridge approach — based on conditioning on diffusions on both ends. This alleviates the need to use the reverse construction used by the earlier papers in this field.

**Summary Of The Review:**

I found the paper hard to read - but found the ideas interesting.

---

> ### Author Response · Authors · 2021-11-18
> **Addressing Issues Raised by Reviewer e93q**
>
> We thank Reviewer e93q for his/her comments, and we address the raised issues.
>
> > In many places, the motivations and constructions are not clear and not written precise enough for the ICLR readers.
>
> In the revised manuscript we reserved $Y$ to denote the time reversal process evolving on noising time, now $X$ always evolves on sampling time.
> The previous ambiguous notation was definitely confusing on this point.
> We additionally revised the manuscript to improve clarity in many places, also following the comments of Reviewer QAeW.
> We added a novel Section 7 to the revised manuscript where we review all the DBMT results and include algorithmic boxes for training and sampling in the DBMT.
> All these changes, jointly with the added toy numerical experiment, make the presentation more clear and precise, and the construction of the proposed method easier to follow.
> Please see the following point addressing the remark on lack of precision.
>
> > The authors need to derive Eq. (7) within the paper for completeness. This result is deferred to several other theorems but it is hard to check whether assumptions made in these works to prove these theorems hold in this setting. What kind of assumptions underlie the original result? Do they hold? / > Also, these results need to be proved clearly instead of referring to the Oksendal’s book as done in the paper.
>
> We do not agree that paper's authors should be required to prove again the results they use.
> Thorough the paper we included many references pointing precisely to the relevant chapters / theorems of standard reference textbooks.
> In the revised manuscript we added for eq. (7) the corresponding theorem in [1], which is freely available in electronic form for personal use.
>
> As explained in the introduction, we relegated the more precise theoretical framework and all the assumptions to Appendix A.
> The results presented in this paper hold for the considered SDE class, i.e. eqs. (11) and (12), with a finite-variance initial distribution $\mathcal{P}_Z$ (see Appendix A).
>
> > I wonder why authors move from Eq. (7) to mixtures, as it is not made clear. Why is it not enough to use Eq. (7) for the generative modelling?
>
> Quoting our work "A diffusion bridge is a diffusion process starting from a given value which is conditioned on hitting a terminal value".
> The diffusion bridge process of eq. (7) always ends at the same terminal value $x_\tau$, so it cannot be used for generative purposes.
>
> > In Eq. (8) where authors introduce the SDE, $g(X_t,t)$ is not specified.
>
> It is defined immediately after eq. (7).
>
> > The authors seem to define in this section an SDE class which are realised through a time-change of simpler SDEs (as they put it), I really couldn’t see how this relates to the rest of the paper.
>
> It gives a better understanding of the role of $\beta_t$ in SDEs (11) and (12), and in turn of the SDEs of [2].
> It represents the instantaneous intensity of time-flow.
>
> > The authors should clarify what they mean by “the following SDEs represent the class of dynamics for (1) and (6)”. If this means that these SDEs, via a change of time argument, can represent (6), this notion has to be made clearer...
>
> Eqs. (1) and (6) are the unconstrained SDEs underlying the approach of [2] and our proposal.
> The complete sentence from our work is "The following SDEs on
> $[0 \tau]$ represent the class of dynamics for (1) and (6) on which we focus in the rest of this paper".
> The "following SDEs", i.e. eqs. (11) and (12), are the specific forms of eqs. (1) and (6) considered from there onward.
> The novel Section 7 in the revised manuscript makes the role of Section 4 clearer.
>
> [1] Särkkä, Solin -- Applied Stochastic Differential Equations
>
> [2] Song, Sohl-Dickstein, Kingma, Kumar, Ermon, Poole -- Score-based Generative Modeling through Stochastic Differential Equations

---

### Official Review · Reviewer_oeyA · 2021-11-02

**Correctness:** 3
**Technical Novelty And Significance:** 3
**Empirical Novelty And Significance:** 2
**Recommendation:** 5
**Confidence:** 4

**Main Review:**

STRENGTHS:

-The paper is well-written. The mixture of diffusion bridges is clearly
motivated and the theoretical results seem to be correct.

-The idea of using diffusion bridges (and mixture of diffusion
bridges) to build a diffusion-based generative modelling is original and
interesting. One of the aspect of this work that I find promising is that in
this setting is that both the terminal and initial distributions are pinned down
contrary to existing works.

-Related to this idea, remarking that a mixture of diffusion remains a diffusion
whose coefficients can be computed is an interesting point to raise and might be
useful in other settings. The fact that the resulting drift can be approximated
using score-matching techniques is also interesting.

WEAKNESSES:

-There is not enough experimental evidence. In particular, even though comparing
the approach of the authors with classical time-reversed methods might be
compute consuming on image synthesis tasks the authors should have provided
comparison on toy examples. As of now it is not clear at all that diffusion
bridge based generative modelling is superior or comparable to time-reversed
diffusion based generative modelling.

-Also, the authors should compared their methods with other works which also pin
down the initial and terminal distributions such as [2,3]

-There is no theoretical justification why diffusion bridge based generative
modelling might be better than the classical time-reversed one. I understand
that the fact that the initial and final distributions are pinned down might help but it
is not clear to me what kind of theoretical result could be obtained.

-Overall I think that the presentation of the paper could be improved if the
authors focused on their core contribution which is the use of mixture of
diffusion bridges for generative modelling. I found Section 4 "SDE Class" and
Section 6 "Transports approximation" to be a bit disconnected from the rest of
the paper. I would have preferred to see more experiments/comparisons on
diffusion bridge models or a more detailed theoretical analysis of these models.

COMMENTS:

-The authors claim that every approach for score-based generative modeling is
 based on time-reversal. I slightly disagree in the sense that [5] never
 explicitly writes down the time-reversal of the forward process but rather find a parametric expression of some backward kernel which is only loosely connected to time-reversal.

-I understand that the authors don't want to clutter the paper with too many
technical results on the existence/uniqueness of SDE and the
existence/positivity of their densities but I think that it would be worth to
check that these assumptions are satisfied in some generative modelling setting.

-I found the experiment presented in Figure 1 to be really interesting but I am
not sure I fully understand what is done here. How do the authors compute the
conditional expectation in Equation (20) and (21)? Is the term in (20) and (21)
computed using score-matching as in [1]? If so, I'm a bit confused at the
finding obtained at the authors. It would seems that the process concentrates on
a single point and therefore that there is no innovation in the generative
modelling process? However it has been observed that score-based generative
modelling is innovative, see Figure 7 in [4] for an ImageNet experiment. Also,
could the authors precise how they compute the weights displayed in Figure 1?

-I think Equation 8 could be slightly simplified to really illustrate the fact that $A(x_t, t)$ is a conditional expectation and therefore can be approximated using a neural network similarly to score-based generative modeling approaches. As of now, the only time this conditional expectation is written is in A.3.2.

[1] Song, Sohl-Dickstein, Kingma, Kumar, Ermon, Poole -- Score-based Generative Modeling through Stochastic Differential Equations

[2] De Bortoli, Thornton, Heng, Doucet -- Diffusion Schrodinger Bridge with Applications to Score-Based Generative Modeling

[3] Vargas, Thodoroff, Lawrence, Lamacraft -- Solving Schrodinger Bridges via Maximum Likelihood

[4] Dhariwal, Nichol -- Diffusion Models Beat GANs on Image Synthesis


**Summary Of The Paper:**

In this paper the authors introduce a new model for diffusion-based generative
modelling.  Instead of relying on time-reversed processes as in existing works,
they propose an approach based on the mixing of diffusion bridges. They show
that a mixing of diffusion bridges remains a diffusion with closed-form drift
and volatility. Similarly to existing score-matching based generative modelling
works the drift in this new formulation can be expressed as a conditional
expectation and therefore can be seen as the minimizer of some loss function,
allowing the use of neural-network based approximations. In addition, the
authors propose some unification of the SDE classes used in [1] and to use
non-identity volatility matrices.

[1] Song, Sohl-Dickstein, Kingma, Kumar, Ermon, Poole -- Score-based Generative Modeling through Stochastic Differential Equations

**Summary Of The Review:**

The main idea of the paper is sound and original. However, I think that this
work might not be mature enough. I would like to see more theoretical and
experimental comparisons with existing works to assess the efficiency of the
proposed method. For these reasons I recommend the rejection of the paper but am
ready to raise my score if the authors provide compelling justifications for
your method.

---

> ### Author Response · Authors · 2021-11-18
> **Addressing Issues Raised by Reviewer oeyA [part 1]**
>
> We thank Reviewer oeyA for his/her comments, and we address the raised issues.
>
> > In this paper the authors introduce a new model for diffusion-based generative modelling. Instead of relying on time-reversed processes as in existing works, they propose an approach based on the mixing of diffusion bridges. They show that a mixing of diffusion bridges remains a diffusion with closed-form drift and volatility.
>
> We apologize, in Section 3.2 we incorrectly defined $\Pi$ to be the law of the solution of (8), but $\Pi$ is the law of the mixture of diffusions.
> This typo is a consequence of a last-minute manuscript update to simplify the notation, we did not notice the introduced problem in time.
> The statement of Theorem 1 is correct (note that $X$ in Theorem 1 is stated to match the marginals, not the law of the mixture of diffusions).
> All the results of the following Sections are unchanged and correct.
> Indeed, the construction of the training objectives of Section 6 heavily rely on the fact that we know how to sample from $\Pi_{t|0,\tau}=P_{t|0,\tau}$, i.e. the transition distribution of the diffusion bridge (if $\Pi$ were the law of (8) the proposed Monte Carlo estimators would not be efficiently implementable).
>
> As a counterexample consider a mixture of Brownian bridges with a "fully dependent" mixing distribution over $(x_0,x_\tau)$ such that for all sampled $(x_0,x_\tau)$ we have $x_0=x_\tau$.
> Then the mixture of diffusions is not Markov (so it cannot be a diffusion): the distribution of $x_\tau|x_t$ for $0<t<\tau$ is different from the distribution of $x_\tau|x_t,x_0=\delta_{x_0}$.
>
> The revised manuscript correct this issue.
> The included toy numerical experiments provide some intuition on the difference between the dynamics of the mixture of diffusions and the process of (8) matching its dynamics.
>
> > There is not enough experimental evidence. In particular, even though comparing the approach of the authors with classical time-reversed methods might be compute consuming on image synthesis tasks the authors should have provided comparison on toy examples. / / I would have preferred to see more experiments/comparisons on diffusion bridge models or a more detailed theoretical analysis of these models. / As of now it is not clear at all that diffusion bridge based generative modelling is superior or comparable to time-reversed diffusion based generative modelling. / There is no theoretical justification why diffusion bridge based generative modelling might be better than the classical time-reversed one. I understand that the fact that the initial and final distributions are pinned down might help but it is not clear to me what kind of theoretical result could be obtained.
>
> We addressed these issues in the thread "Addressing Key Issues".
>
> >Also, the authors should compared their methods with other works which also pin down the initial and terminal distributions such as [2,3]
>
> We added comparison considerations in the Related Work section of Appendix C.
>
> > Overall I think that the presentation of the paper could be improved if the authors focused on their core contribution which is the use of mixture of diffusion bridges for generative modelling.
>
> The revised manuscript shortens the section on random fields for space modelling, most of which is now in the Appendix, to focus more on the DBMT proposal.
>
> > I found Section 4 "SDE Class" and Section 6 "Transports approximation" to be a bit disconnected from the rest of the paper.
>
> Section 4 is needed to implement the training objectives of Section 6.
> We understand that, lacking any training experiment, the results of Section 6 feels disconnected.
> However, they are central to the proposed methodology.
> The novel Section 7 in the revised manuscript, with the corresponding algorithm box for training, makes the relevance of Section 6 clearer.
>
> > The authors claim that every approach for score-based generative modeling is based on time-reversal. I slightly disagree in the sense that [5] never explicitly writes down the time-reversal of the forward process but rather find a parametric expression of some backward kernel which is only loosely connected to time-reversal.
>
> We are very interested in [5] and would like to comment further, but it appears that the reference has been cut from the review.
> Could Reviewer oeyA please provide the reference for [5] with a further comment?
>
> > I understand that the authors don't want to clutter the paper with too many technical results on the existence/uniqueness of SDE and the existence/positivity of their densities but I think that it would be worth to check that these assumptions are satisfied in some generative modelling setting.
>
> They are satisfied for the SDEs of eq. (11), (12), which is the SDE class considered in this paper, when $\mathcal{P}_Z$ has finite variance (see Appendix A).

---

> > ### Author Response · Authors · 2021-11-18
> > **Addressing Issues Raised by Reviewer oeyA [part 2]**
> >
> > > I found the experiment presented in Figure 1 to be really interesting but I am not sure I fully understand what is done here. How do the authors compute the conditional expectation in Equation (20) and (21)? Is the term in (20) and (21) computed using score-matching as in [1]?
> >
> > We provided an expanded description in the thread "Addressing Key Issues", please let us know if this explanation is satisfactory.
> >
> > > If so, I'm a bit confused at the finding obtained at the authors. It would seems that the process concentrates on a single point and therefore that there is no innovation in the generative modelling process? However it has been observed that score-based generative modelling is innovative, see Figure 7 in [4] for an ImageNet experiment. Also, could the authors precise how they compute the weights displayed in Figure 1?
> >
> > What can be inferred from Figure 1 is that the terminal sample gets "decided" with high probability well in advance of terminal time.
> > That is, evaluating $\mathbb{E}[X_0]$ for both the *true score* and the *trained model score* at any $t \geq 0.6$ gives a very strong indication of what the terminal sample will be (for the *trained model score* some details are missing, which is not the case for the *true score*).
> > However, the top of Figure 1, with associated first 2 cells' rows, only depicts a single trajectory (the remaining part concerns 2 additional trajectories: one for rows 3 and 4, one for rows 5 and 6).
> > We addressed this lack of clarity in Figure 1's caption in the revised manuscript.
> > To conclude, there is diversity across trajectories (generations), but there is a concentration phenomenon over sampling time for each trajectory.
> >
> > > I think Equation 8 could be slightly simplified to really illustrate the fact that  is a conditional expectation and therefore can be approximated using a neural network similarly to score-based generative modeling approaches. As of now, the only time this conditional expectation is written is in A.3.2.
> >
> > We kept eq. (8) as it is for space reasons, but added the suggested representation in the following paragraph in the revised manuscript.
> >
> > > The main idea of the paper is sound and original. However, I think that this work might not be mature enough. I would like to see more theoretical and experimental comparisons with existing works to assess the efficiency of the proposed method. For these reasons I recommend the rejection of the paper but am ready to raise my score if the authors provide compelling justifications for your method.
> >
> > Please see the thread "Addressing Key Issues" where we address the lack of benchmarking and the benefits of the proposed approach.
> > The novel Section 7 in the revised manuscript makes the simplicity of the proposed approach more evident.

---

> > > ### Comment · Reviewer_oeyA · 2021-11-28
> > > **Response to rebuttal + decision**
> > >
> > > I have read the other reviews and the author rebuttal.  I thank the authors for
> > > their explanations/clarifications.  I still have some concerns with this paper
> > > which I will try to develop further.
> > >
> > > I still find the experiment reported in Figure 1 to be hard to read even though
> > > the clarifications provided by the authors in the revised version of the paper
> > > and the rebuttal are useful. I think I understand how the weights are computed
> > > but an explicit formula (even in the appendix) would be much useful. Also, I'm
> > > still confused because the fact that one weight degenerates to one means that we
> > > recover one sample from the original dataset. However, in practice generative
> > > modelling focuses on synthesizing new samples from the underlying data
> > > distribution and it has been shown that score-based generative modelling is able
> > > to produce new samples (see my original review). Maybe this is due to the fact that the score is computed
> > > exactly here by looping through the whole dataset but I find this confusing. I
> > > didn't find the answer provided by the authors regarding this problem to be very
> > > satisfying: "To conclude, there is diversity across trajectories (generations),
> > > but there is a concentration phenomenon over sampling time for each
> > > trajectory.". Finally, and more importantly, even though I think the findings
> > > of Figure 1 are interesting they do not motivate the Diffusion Bridge Mixture
> > > Transport (DBMT) approach. It is hard to see how this experiment fits in the
> > > whole paper even after a careful reading of the second version of the paper.
> > >
> > > My second main concern is the motivation of the paper which remains quite
> > > unclear to me. Reading your "Additional clarification [part 1]'" answer to
> > > Reviewer QAeW I understand that one of the main appeal of the DBMT approach
> > > would be that "there is no requirement to have a large time interval''. I agree
> > > that since DBMT relies on bridge this is true in theory and seems to be one
> > > advantage of the method. However, there might be some practical limitations and
> > > one cannot make the final time $\tau$ arbitrary small in practice while keeping
> > > a fixed stepsize in the discretization (otherwise only step of the method would
> > > be able to produce data points from a Gaussian sample). One key experiment (even in a toyish setting) that is missing is a clear comparison between DBMT and traditional score-matching approaches. I'm not really convinced
> > > by the argument regarding the freedom of the choice of the coefficients $f$ and
> > > $g$ that the authors provide in ``Addressing Key Issues [part 2]'' because in
> > > practice $f$ is linear in $x$ and $g$ is independent from $x$.
> > >
> > > To summarize, even though the authors have clarified their approach and one of
> > > the experiment I still think that some work remains to be done to provide a
> > > clearer picture to the reader.
> > >
> > > Other remark:
> > >
> > > -Thanks for the clarification regarding the distribution of the process.
> > > I also appreciate the theoretical clarifications provided by the authors.
> > >
> > > -Sorry about the missing reference [5]. I was referring to:
> > > ``Denoising diffusion probabilistic models'' (Ho, Jain, Abbeel) In this paper,
> > > even though a backward kernel is identified. It is not explicit that the
> > > obtained backward kernel corresponds to some time-reversal.
> > >
> > > -I find the new version of the conclusion to be hard to read. I think it is more
> > >  confusing than the previous version. Some important sentences seem to have been
> > >  cut.

---

> > > > ### Author Response · Authors · 2021-11-29
> > > > **Reply**
> > > >
> > > > > I still find the experiment reported in Figure 1 to be hard to read [...] I think I understand how the weights are computed but an explicit formula (even in the appendix) would be much useful.
> > > >
> > > > We are unable to revise the manuscript as we are past the allowed rebuttal time.
> > > > We can add this formula to the Appendix of the camera ready version.
> > > >
> > > > > Also, I'm still confused because the fact that one weight degenerates to one means that we recover one sample from the original dataset [...]
> > > >
> > > > In the no-approximations / infinite-capacity limit (i.e. *true score* -- top of part of Figure 1 being discussed here) $\mathcal{P}_\mathcal{D}$ is replicated exactly. $\mathcal{P}_\mathcal{D}$ is the empirical distribution of the training data.
> > > > This is a general result about generative models based on approximators with unbounded representation power (i.e. neural networks).
> > > > It would be very surprising if it were otherwise.
> > > > It holds for our work, for the work of Song et al., and for all related works referenced in our paper and in this discussion.
> > > > See "Rezende, Danilo Jimenez, and Fabio Viola -- Taming VAEs" for yet the same result in a completely different context (specifically, first paragraph following Proposition 2).
> > > >
> > > > > I didn't find the answer provided by the authors regarding this problem to be very satisfying: "To conclude, there is diversity across trajectories (generations), but there is a concentration phenomenon over sampling time for each trajectory.".
> > > >
> > > > Our quoted sentence is **not** an answer to the remark raised above.
> > > > It is a clarification to a previous misunderstanding: 1 trajectory is only representative of 1 sample.
> > > > And there is no "problem" specific to our approach (compared to any other neural-network based generative approach), as we remarked in the previous comment.
> > > >
> > > > > Finally, and more importantly, even though I think the findings of Figure 1 are interesting they do not motivate the Diffusion Bridge Mixture Transport (DBMT) approach [...]
> > > >
> > > > Figure 1 shows the properties of the target mapping being approximated by NNs.
> > > > It also shows some shortcomings of time-reversal approaches.
> > > > It is not used to motivate the DBMT approach, and we do not claim so in the paper.
> > > >
> > > > > My second main concern is the motivation of the paper which remains quite unclear to me. Reading your "Additional clarification [part 1]'" answer to Reviewer QAeW I understand that one of the main appeal of the DBMT approach would be that "there is no requirement to have a large time interval''. I agree that since DBMT relies on bridge this is true in theory and seems to be one advantage of the method. However, there might be some practical limitations and one cannot make the final time $\tau$ arbitrary small in practice while keeping a fixed stepsize in the discretization (otherwise only step of the method would be able to produce data points from a Gaussian sample). One key experiment (even in a toyish setting) that is missing is a clear comparison between DBMT and traditional score-matching approaches.
> > > >
> > > > A comparison would be based on a fixed number time-steps $T$, i.e. same computational budget, not on a fixed time-step interval (as moreover time and SDEs can be jointly rescaled to yield equivalent dynamics, time here does not correspond to any physical quantity).
> > > > We argue that experimental results this kind are application-dependent, but in any case there is no scope to add further experiments at this stage.
> > > >
> > > > > I'm not really convinced by the argument regarding the freedom of the choice of the coefficients $f$ and $g$ that the authors provide in ``Addressing Key Issues [part 2]'' because in practice $f$ is linear in $x$ and $g$ is independent from $x$.
> > > >
> > > > The class of SDEs that we consider is more general than the one considered in Song et al. and in any of the other works referenced in the manuscript and in this discussion.
> > > > No prior work considered non-factorial transitions due to complexity considerations.
> > > > But this not really the main point.
> > > > The key issue is that to achieve ergodic behavior in time-reversal approaches on a fixed time interval it is necessary to formulate $f$ and $g$ in a certain way (i.e. the large $\beta_t$ in VESDE and VPSDE).
> > > > This constraint does not apply to the DBMT.
> > > >
> > > > > Sorry about the missing reference [5]. I was referring to: ``Denoising diffusion probabilistic models'' (Ho, Jain, Abbeel) In this paper, even though a backward kernel is identified. It is not explicit that the obtained backward kernel corresponds to some time-reversal.
> > > >
> > > > This connection is established in the work of Song et al.
> > > >
> > > > > I find the new version of the conclusion to be hard to read. I think it is more confusing than the previous version. Some important sentences seem to have been cut.
> > > >
> > > > We moved a good part of text from the Conclusion Section to Section 7.
> > > > As the rebuttal does not allow for additional pages, and the paper is already quite compressed, it will be difficult to improve further on this point.

---

### Official Review · Reviewer_QAeW · 2021-11-02

**Correctness:** 4
**Technical Novelty And Significance:** 3
**Empirical Novelty And Significance:** 1
**Recommendation:** 3
**Confidence:** 3

**Main Review:**

Post-response update. The authors' addressed my technical comments, and revised the manuscript
with clarifying visualisations and algorithm boxes. The three main major issues with the paper nevertheless still stand: the presentation is overly complex; the motivation is not convincing enough; and experiments are anecdotal.

------------

This is a theory paper with only minimal experiments. The paper analyses Song et al ’21 bi-directional diffusion model, and proposes to replace and/or generalise it with Brownian bridge formulation, which only moves in generative direction. The problem is somewhat poorly motivated and it is not entirely clear how significant are the problems with the bidirectional approach, or what are the advantages of the bridge. One would expect that the bridge formulation now has a major problem with the (x0,xtau) pairing: surely a poor pairing would lead to a poor model.

The paper derives the bridge formulation at length. The derivation of the method takes almost the whole paper, and I found it to be difficult to follow. The paper does a poor job of describing what is the final model (algorithm box would help). The paper is also filled with tons of math, but has little exposition or motivation on the higher level concepts or intuition. There are also almost no illustrations of the ideas. Ultimately, after several hours on this paper, I only could understand around half of the material. The reader is overburdened by notation. This is going to limit the paper to narrower audience and impact. The scope and contributions are not sufficiently clear.

The paper shows that Song’21 method can be seen as special cases of covariant Brownian and OU processes. This seems like a side-result and not particularly useful, but is nevertheless interesting. The paper also presents some discussion of spatialization of the generative process with Gaussian processes. This is again interesting, but feels incomplete or more of a sketch. There are only very tentative results regarding this.

There are no large-scale or exhaustive experiments. THis is somewhat surprising: the proposed method seems like a variant of Song’s method, and I wonder why no comparative experiments were done. Perhaps the authors run out of time. I feel that the theoretical results alone are not sufficient for publication: from SDE perspective changing a reversible SDE to Brownian Bridge is rather incremental, and thus experimental demonstration would have been necessary to make the paper strong enough for publication.


Technical comments
* It seems that “terminal distribution” is at time 0, while “initial distribution” is at time tau. Yet, in (i) in 1st paragraph the initial time is 0, and this is for observed objects. In (1) in 3rd paragraph the time 0 for latent base object. These are confusing, and needs clarification. A figure would be helpful.
* Sec 1 “Firstly..”. “Q_tau retains dependency on P_D”. This could be non-true: if forward process mixes, then it will become oblivious of the initial distribution. This is the goal of density destructors: to lose all information of the original distribution. “The exact reverse-time simulation..”. I can’t understand this sentence. Why are we starting from Q_tau, and not Q_0? I also don’t see what approximation does this “firstly” subparagraph talk about, what was the problem again? Try to clarify
* Secondly: why is it a problem that neural network approximates diffusion process? Is the problem that it introduces bias, or that it has some residual? Don’t we always need to approximate the diffusion if we don’t have access to the true one? Text also claims we have known reverse time dynamics known, surely this can’t be true? If we know the reverse time dynamics exactly, then why are we learning it? Try to clarify the problem and motivation of this point.
* Thirdly: this is very clear
* What is x0? This has not been defined, so I don’t understand if this is supposed to be a real object, a random noise object from base distribution, or something in else. It also seems that in 3rd paragraph the forward direction is now generative direction, while previously it was destructive direction. Please clarify to reader.
* I generally can’t follow the 3rd paragraph. The text talks about DDPM, DBMT, and redefines DDPM as DTRT. There is lots of detail, but not enough background to understand. It seems the main problem is that Q_tau depends on P_D, but it is unclear why is this a problem.
* In 4th paragraph the “drift adjustment” is unexplained. It is also implied that it is bad that the adjustment depends on data. Surely everything depends on data? I fail to see what is the argument here
* The f/g are discussed, but not explained
* Sec 2. “u” is often used in PDEs to denote solutions, but here its time. This can introduce unnecessary confusion for some readers.
* I can’t follow what transition density q_u|u’(y|x) means. y is undefined (so is actually x as well). What does transition density mean? Is it a conditional marginal distribution or something else? Please define
* Before eq 2 we first define q_u to be marginal density, while then eq2 redefines the density as a finite mixture. Surely marginal density should refer to "fully" continuous density, while eq2 should then be an approximation of the marginal density, or the N-empirical marginal density?
* In eq 3 we define X_0 ~ P_Z, while in eq 1 we define X_0 ~ P_D. This is confusing
* Eq 4 is confusing. The first line seems to be a mixture density, but second line Q_0,u is undefined. Does htis equation just state that one can do MC to sample from a mixture density? What is dX_u?
* Eq 8: how does one do the pairing between x0 and xtau^{(n)}, or is there any pairing?
* eq 14: \Gamma looks like a function here, is it?
* eq 21: weights \omega are introduced, and it seems that they are central to the method. Yet, they are not part of any of the equations. They need introduction and needs to be connected to the presented theory
* Sec 4. I got lost here. There is tons of math and theory here, which looks simple enough to follow, but little of this is motivated and the reader gets lost on where are we going with this. The presentation remains on a very technical level, not exposing the underlying concepts and intuition. Not all math needs to be understandable by a reader, but even a casual reader should understand why the math is there or what it achieves. Here I don’t see why the Song’s method is reinterpreted as the two processes 9+10. Does this relate to the rest of the paper or the proposed method? The presentation is also difficult to follow since a lot is left unexplained intentionally (eqs 15..17). The authors need to describe the motivation and role of the math, and also need to conceptualise the theory by helpful descriptions (or figures).
* Sec 5. I’m again lost here. So eq 14 is applied to 8 to get 20. But 14 applies to Brownian and OU processes, while eq 8 is a bridge process. Why would the result hold? Furthermore, eq 8 gives a quite complex mixture model, which again seems very different from eq 9 or 10. It seems odd that to derive a mixture model 8, we take a detour to a different (and very simple) model family. Can’t we derive the eq 14 directly from eq 8?
* Furthermore, the full method proposed in the paper is difficult to follow, since its description is scattered in many places. It’s difficult to go back-and-forth between eq 8 and 20. Later one needs to go back-and-forth between eq 23, 20 and 8 again. The description is also not complete yet, since \Pi, v(), a(),etc are not defined. Because of these challenges, I then couldn’t follow the paragraphs after eq 21.
* Where is the \omega coming from? What is Y? What is E[Y]?}) It seems that method becomes some kind of weight-tuning algorithm, but weights are completely absent from any equations. Including an algorithm block would be helpful.
* It seems that E[Y_t|x,t] becomes a central term later in the paper, but they are not properly defined. In which way they refer to the expectation terms in 20+21? Please define.
* It would be very helpful to give the \sum_n=1^N version of eq 8, since it seems that the proposed theory in the paper is based on the mixture view (or is it?)
* Figure1: Generally I don’t understand what this experiment is doing. I can’t understand what happens on top. What is “all samples x^{(n)} in CIFAR”? This should be 50k points I assume, but it seems we only visualise around 10 curves. What does “true score” mean? How does one get the “true” score, and what does the score mean? What does Euler(1000) mean? What are the colors? Also at bottom I can’t understand what is "true score”, and how come the score is an image, it was said to be a “single weight” above, so one expects it to be a scalar? The caption says that we track scores, but these look like images instead. Second and third rows are trained models, but what is then first row (is it untrained?). If we follow here E[Y], then should these images be some kind of average trajectories? The text describes three stages, but I can’t see where these come from (or what they are). I’m again having lots of trouble understanding the fig1 explanations in sec 5 since it’s difficult to follow what the figure shows.
* Eq 25: why would matching s≈X result in s≈E[Y]? These seem like very different targets.


**Summary Of The Paper:**

The paper studies the Brownian bridge formulation for diffusion-based generative models, derives theory for it, and shows connections to earlier works. Some spatial developments are also discussed. This is a theoretical paper that only has rudimentary experiments. The overall idea of Brownian Bridge diffusion is interesting, but the idea’s merits are not shown. The paper suffers from high technical complexity making it difficult to digest.


**Summary Of The Review:**

While the theoretical derivations and the idea of Brownian bridge are interesting, and potentially a breakthrough, I feel that the theoretical results alone are not sufficient for publication. Changing the reversible SDE to BB is still somewhat incremental proposal, and thus the experimental demonstration would have been necessary to make the paper strong enough for publication. Furthermore, the presentation of the paper not good enough for a publication.

---

> ### Author Response · Authors · 2021-11-18
> **Addressing Issues Raised by Reviewer QAeW [part 1]**
>
> We thank Reviewer QAeW for his/her comments, and we address the raised issues.
>
> > The paper analyses Song et al ’21 [...] it is not entirely clear how significant are the problems with the bidirectional approach, or what are the advantages of the bridge. One would expect that the bridge formulation now has a major problem with the (x0,xtau) pairing: surely a poor pairing would lead to a poor model.
>
> A good pairing requires less work from the process $X$ to transport $\mathcal{P}_Z$ to $\mathcal{P}_\mathcal{D}$.
> But in [1] the pairing cannot be particularly helpful because a strong linkage between $\mathcal{P}_Z$ and $\mathcal{P}_\mathcal{D}$ corresponds to a large approximation error of $Q_\tau$ by $\mathcal{P}_Z$.
> In this sense, we see the possibility of achieving a good pairing as a beneficial point of the proposed approach.
> The proposed approach has more flexibility in the choice of $\mathcal{P}_Z$, as it is disentangled from $f()$ and $g()$.
>
> > The paper does a poor job of describing what is the final model (algorithm box would help) / The presentation is also difficult to follow since a lot is left unexplained intentionally (eqs 15..17). / Furthermore, the full method proposed in the paper is difficult to follow, since its description is scattered in many places [...]
>
> We added a novel Section 7 to the revised manuscript where we review all the DBMT results and include algorithmic boxes for training and sampling in the DBMT.
>
> > There are no large-scale or exhaustive experiments [...]
>
> We commented on this point in the thread "Addressing Key Issues".
>
> > It seems that “terminal distribution” is at time 0, while “initial distribution” is at time tau. Yet, in (i) in 1st paragraph the initial time is 0, and this is for observed objects. In (1) in 3rd paragraph the time 0 for latent base object. These are confusing [...] / “The exact reverse-time simulation..”. I can’t understand this sentence. Why are we starting from Q_tau, and not Q_0? / It also seems that in 3rd paragraph the forward direction is now generative direction, while previously it was destructive direction. Please clarify to reader. / In eq 3 we define X_0 ~ P_Z, while in eq 1 we define X_0 ~ P_D. This is confusing
>
> We agree that this confusing.
> The confusion is caused by the overloading of both noising and sampling time processes under the same notation (we tried to simplify the notation and relied on $t$ vs $u$ to discriminate between the sampling and the noising process, but went too far).
> In the revised manuscript we reserved $Y$ to denote the process evolving on over time, now every process $X$ evolves over sampling time.
> This especially made Section 5 clearer.
>
> > Sec 1 “Firstly..”. “Q_tau retains dependency on P_D”. This could be non-true: if forward process mixes, then it will become oblivious of the initial distribution. This is the goal of density destructors: to lose all information of the original distribution / I also don’t see what approximation does this “firstly” subparagraph talk about, what was the problem again? / It seems the main problem is that Q_tau depends on P_D, but it is unclear why is this a problem.
>
> The first approximation (i.e. in "Firstly...") is the approximation due to evolving the noising process on a finite time internal.
> The refereed theorem of [2] provides a quantitative measure for the impact of this approximation.
> Perfect mixing requires infinite time, and our approach avoids the need for this approximation.
>
> > Secondly: why is it a problem that neural network approximates diffusion process? Is the problem that it introduces bias, or that it has some residual? Don’t we always need to approximate the diffusion if we don’t have access to the true one? Text also claims we have known reverse time dynamics known, surely this can’t be true? If we know the reverse time dynamics exactly, then why are we learning it? Try to clarify the problem and motivation of this point.
>
> The reverse, i.e. sampling, time dynamics are known, and given in eq. (3).
> But the computation of $\nabla_y \ln q_r(y)$ from (2) requires traversing the whole training dataset at each discretization step to simulate from (3) (we do this to produce Figure 1 (top)).
> Because of this, and because the goal is not to replicate the training dataset, a neural network approximation is used.
>
> > What is x0? This has not been defined, so I don’t understand if this is supposed to be a real object, a random noise object from base distribution, or something in else.
>
> It is a generic value, we clarified this in the revised manuscript.
>
> > I generally can’t follow the 3rd paragraph. The text talks about DDPM, DBMT, and redefines DDPM as DTRT.
>
> We used DTRT specifically to indicate the diffusion based continuous-time variant of DDPM, the revised manuscript is clearer on this point.

---

> > ### Author Response · Authors · 2021-11-18
> > **Addressing Issues Raised by Reviewer QAeW [part 2]**
> >
> > > In 4th paragraph the “drift adjustment” is unexplained. It is also implied that it is bad that the adjustment depends on data. Surely everything depends on data? I fail to see what is the argument here
> >
> > We did not imply that it is bad that it depends on the data.
> > The issue is the associated computational complexity: each point evaluation requires traversing the whole dataset (see reply above).
> > This motivates the whole developments of Section 6, where neural network approximations to the drift adjustment are introduced.
> >
> > > The f/g are discussed, but not explained
> >
> > They are the coefficient of SDE (1), this is explained in Section 2.
> >
> > > Sec 2. “u” is often used in PDEs to denote solutions, but here its time. This can introduce unnecessary confusion for some readers.
> >
> > The revised manuscript uses $r$, which is a good mnemonic for "remaining sampling time".
> >
> > > I can’t follow what transition density q_u|u’(y|x) means. y is undefined (so is actually x as well). What does transition density mean?
> >
> > A transition density of a stochastic process is for example $p_{t|s}(x_t|x_s)$ for $s<t$.
> > It is the conditional probability density of a state at a later time ($x_t$) given the current state ($x_s$).
> > The variables $x,y$ in $q_u|u’(y|x)$ are muted, or local, they do not refer to objects defined anywhere else.
> > We added an explanation to the notation paragraph.
> >
> > > Before eq 2 we first define q_u to be marginal density, while then eq2 redefines the density as a finite mixture.
> >
> > It is always a density, because (2) is a mixture of conditional densities.
> > Eq. (2) is only making explicit the form of $q_u$.
> >
> > > Eq 4 is confusing. The first line seems to be a mixture density, but second line Q_0,u is undefined. Does this equation just state that one can do MC to sample from a mixture density? What is dX_u?
> >
> > Q is the law of the process $X$ over noising time, as defined just after eq. (1).
> > Q_{0,u} is the joint distribution of this process at times $(0,u)$.
> > We rely on the notation explained in the notation paragraph (part on finite dimensional distributions).
> > $dX_u$ is used as muted variable for the probability distribution (it would not make sense to use $x$ as in densities, as distributions are not evaluated in a point-wise way).
> > We added explanations to the notation paragraph.
> >
> > > Eq 8: how does one do the pairing between x0 and xtau^{(n)}, or is there any pairing?
> >
> > Some alternatives and with related considerations are discussed in the Conclusion section (or in Section 7 of the revised manuscript).
> >
> > > eq 14: \Gamma looks like a function here, is it?
> >
> > No, it is always a constant matrix, as previously defined.
> > In many formulas there are quantities involving a matrix times a scalar, we decided to consistently keep the matrix as the first term.
> >
> > > eq 21: weights \omega are introduced, and it seems that they are central to the method. Yet, they are not part of any of the equations.
> >
> > Section 5 is not developing the proposed method.
> > It exploits representation results that we derived (eqs. (20), (21)) to yield insights on the workings of [1].
> > See the thread "Addressing Key Issues".
> >
> > > Sec 4. I got lost here [...] I don’t see why the Song’s method is reinterpreted as the two processes 9+10. Does this relate to the rest of the paper or the proposed method?
> >
> > The VESDE and VPSDE of [1] are shown to be equivalent to the $\beta_t$-time-changes of (18), (19).
> > This clarifies the role of $\beta_t$ in VESDE and VPSDE (which is the only SDE parameter in [1]) as instantaneous time-flow.
> >
> > > Sec 5. I’m again lost here. So eq 14 is applied to 8 to get 20. But 14 applies to Brownian and OU processes, while eq 8 is a bridge process. Why would the result hold?
> >
> > Eq. (8) is not a bridge process, it is the SDE of the process realizing the proposed transport.
> > We plug in the specific form of (14) for the SDE class of interest to get (20) after some calculations (Appendix A).
> > > Furthermore, eq 8 gives a quite complex mixture model, which again seems very different from eq 9 or 10. It seems odd that to derive a mixture model 8, we take a detour to a different (and very simple) model family. Can’t we derive the eq 14 directly from eq 8?
> >
> > The processes $Z$ in Section 4, i.e. eq (9), (10), are only used to construct the processes $X$ of eq. (11), (12), which are the SDE class that we consider in this work.
> > Equations (13) to (17) report analytical formulas for $X$ following (11), (12).
> > These formulas are needed to implement the training objectives of Section 6.
> > Section 7 of the revised manuscript makes the DBMT construction clearer.

---

> > > ### Author Response · Authors · 2021-11-18
> > > **Addressing Issues Raised by Reviewer QAeW [part 3]**
> > >
> > > > Where is the \omega coming from? What is Y? What is E[Y]?}) It seems that method becomes some kind of weight-tuning algorithm, but weights are completely absent from any equations. Including an algorithm block would be helpful. / It seems that E[Y_t|x,t] becomes a central term later in the paper, but they are not properly defined. In which way they refer to the expectation terms in 20+21? Please define.
> > >
> > > As previously noted, Section 5 is not developing the proposed method.
> > > Please see the thread "Addressing Key Issues" for a detailed explanation of Section 5.
> > > We modified Section 5 in the revised manuscript to make the presentation clearer, also thanks to the revised notation (introduction of process $Y$, see point above).
> > >
> > > > It would be very helpful to give the \sum_n=1^N version of eq 8, since it seems that the proposed theory in the paper is based on the mixture view (or is it?)
> > >
> > > Due to space constraints we decided not to add this extra equation.
> > > Moreover, the drift of eq. (8) as written is already a mixture (mixtures do not have to be over a finite number of components).
> > >
> > > > Figure1: Generally I don’t understand what this experiment is doing. I can’t understand what happens on top. What is “all samples x^{(n)} in CIFAR”? This should be 50k points I assume, but it seems we only visualise around 10 curves. What does “true score” mean? How does one get the “true” score, and what does the score mean? What does Euler(1000) mean? What are the colors? Also at bottom I can’t understand what is "true score”, and how come the score is an image, it was said to be a “single weight” above, so one expects it to be a scalar? The caption says that we track scores, but these look like images instead. Second and third rows are trained models, but what is then first row (is it untrained?). If we follow here E[Y], then should these images be some kind of average trajectories? The text describes three stages, but I can’t see where these come from (or what they are). I’m again having lots of trouble understanding the fig1 explanations in sec 5 since it’s difficult to follow what the figure shows.
> > >
> > > We provided an expanded description in the thread "Addressing Key Issues".
> > > We now address the additional remarks: Euler(T) is the Euler discretization scheme for SDEs with T integration steps, this is explained just before eq. (22); we do not visualize scores: the top of Figure 1 visualizes the weights $\omega(x_t,t)^{(n)}$, the rest of Figure 1 visualizes $\mathbb{E}[X_0]$ (current expected terminal sample) and $X_t$ (current evolving sample), we understand the confusion due to ambiguous wording: in "evolution $\omega(x_t,t)^{(n)}$ for..." what follows "for..." defines the case for which $\omega(x_t,t)^{(n)}$ is shown, not additional quantities being shown, we improved the wording in the revised manuscript; all expectations are conditional to a given time and space point of a given trajectory, there is no averaging across trajectories; the tree stages are the temporal stages $0.4 < t$, $0.4 \leq t \leq 0.6$, $t > 0.6$ associated to the generation of a single sample, they are associated to different dynamics of $\omega(x_t,t)^{(n)}$ over time.
> > > Please let us know if any point remains unclear.
> > >
> > > > Eq 25: why would matching s≈X result in s≈E[Y]? These seem like very different targets.
> > >
> > > The function minimizer $s_\phi(x,t)$ of (25) is $\mathbb{E}[X_\tau|x_t]$, where the expectation is under the law of the diffusion bridge mixture, i.e. $\Pi$.
> > >
> > > > While the theoretical derivations and the idea of Brownian bridge are interesting, and potentially a breakthrough, I feel that the theoretical results alone are not sufficient for publication. Changing the reversible SDE to BB is still somewhat incremental proposal, and thus the experimental demonstration would have been necessary to make the paper strong enough for publication. Furthermore, the presentation of the paper not good enough for a publication.
> > >
> > > We addressed the lack of clarity with numerous changes in the manuscript as detailed in the previous points and in the thread "Addressing Key Issues".
> > > We added a simple numerical experiment to make the proposal more concrete and to support the validity of the proposed method.
> > >
> > > [1] Song, Sohl-Dickstein, Kingma, Kumar, Ermon, Poole -- Score-based Generative Modeling through Stochastic Differential Equations
> > >
> > > [2] De Bortoli, Thornton, Heng, Doucet -- Diffusion Schrödinger Bridge with Applications to Score-Based Generative Modeling

---

### Official Review · Reviewer_FUzA · 2021-11-03

**Correctness:** 4
**Technical Novelty And Significance:** 4
**Empirical Novelty And Significance:** 1
**Recommendation:** 8
**Confidence:** 2

**Main Review:**

The methods presented are clear improvements to existing literature relaxing the constraints of time-reversal diffusions and moving beyond fully factorial noise models. The flow of the writing is clear, supporting arguments are properly presented, and representative results are shown appropriately. In addition, a code package is provided by the authors in the comments for reproducibility purposes which is clean and well organized. Overall, I did not find specific issues in this paper and hence I do recommend this paper for publication. That said, I'm not an expert in this field and hence I'll leave some room for my lack of knowledge about the state-of-the-art and significance of the work.

**Summary Of The Paper:**

The paper introduces a methodological framework for generative modeling through diffusion processes without time-reversal arguments. By utilizing diffusion bridges the authors consider processes that start and end in pre-determined points $x_0$ and $a_{\tau}$. Then by extending this to mixtures of diffusion bridges they show how to transport a distribution $\Pi_0 = P_D$ to $\Pi_{\tau} = P_Z$ without time-reversal of the diffusion process. For specific classes of SDEs by drawing connections to the paradigm introduced by Song et al 2021 and provide a unified view on the drift adjustment in forward and backward SDEs. This allows for defining a time and state-dependent class probability function (given by conditional expectation) providing further insight into the inner-workings of DTRT and DBMT. The authors then introduce two other training objectives $L_{\text{FD}},L_{\text{CE}}$ and discuss their favorable properties. Finally, methods for moving beyond fully factorial noise models are presented by extending the process and the noise term from functions of time to be functions of both time and space. This allows for incorporating "priors" with better spatial dependency models more suitable for specific datasets.

**Summary Of The Review:**

Writing is clear, problem is significant, and proposed methods are novel. I did not find specific issues in this paper (although I did not go through the details of appendix A).

---

> ### Author Response · Authors · 2021-11-22
> **Review reply**
>
> We thank Reviewer FUzA for his/her review and we remain available for clarifications if needed.

---

### Author Response · Authors · 2021-11-10
**Addressing Key Issues**

We start by addressing the issues raised by multiple Reviewers that are central to our rebuttal.
We separately addressed the remaining issues raised specifically by each Reviewer.

-- comments updated with uploaded revised paper --

### Clarifying Figure 1 and Related Experiment

Reviewers QAeW and oeyA find the numerical experiment of Section 5, and associated Figure 1, unclear and difficult to understand.
We provide below a detailed explanation, and we have updated the manuscript to make Section 5 easier to follow.
The explanation below uses to the notation of the original submission, in the revised manuscript we identify with $Y$ the time reversal process.

To start with, consider the VPSDE diffusion of eq. (19) which determines the transition density $q_{u|0}(x_{u}|x_0)$ (the description of the time-varying parameters $\beta_{\mathrm{vp},u},\beta_{\mathrm{ve},u}$ and the form of the transition density can be found in Section 4 supplemented by Appendix B, or in the original work of [1]).
The transition density in turn determines the marginal density $q_u(x_u)$ via eq. (2) where here $x^{(1)},\dots,x^{(N)}$ are the 50000 samples of the training portion of CIFAR10, i.e. $\mathcal{D}(\text{cifar})$.
The *true score* mentioned in Figure 1 refers to the gradient of the logarithm of $q_u(x_u)$, i.e. $\nabla_{x_u} \ln q_u(x_u)$.
Each pointwise evaluation of $\nabla_{x_u} \ln q_u(x_u)$ requires traversing $\mathcal{D}(\text{cifar})$.

From Section 5, we know that the evolution of the VPSDE diffusion $X_t$ over sampling time is driven by the drift-adjustment term of eq. (21).
As sampling time progresses, $X_t$ is pulled toward the expectation term $\mathbb{E}[X_0]$ of eq. (21) with increasing strength due to the scalar functions $a,v$: $a$ goes to 1, $v$ goes to 0 (the exact forms of $a,v$ are reported in Appendix B).
$\mathbb{E}[X_0]$ is given by $\mathbb{E}[X_0] = \sum_{n=1}^N x^{(n)} \omega(x_t,t)^{(n)} = \sum_{n=1}^N x^{(n)}q_{0|t}(x^{(n)}|x_t)$.
The weight $\omega(x_t,t)^{(n)}$ for the sample $x^{(n)}$ is the probability, under the VPSDE model, or reaching $x^{(n)}$ at terminal time given that we are at time $t$ with position $X_t=x_t$.
The top part of Figure 1 plots the weights $\omega(x_t,t)^{(n)}$ for all samples in $\mathcal{D}(\text{cifar})$ over a single simulated trajectory of the VESDE model over sampling time.
The color palette is cyclical (colors are used only to distinguish different lines), and only a few lines are visible because most of the weights $\omega(x_t,t)^{(n)}$ are close to 0 over the whole sampled trajectory.
The first and second rows of Figure 1 show respectively the evolution of $\mathbb{E}[X_0]$ and that of $X_t$ for this same trajectory.
It can be seen that changes in $\mathbb{E}[X_0]$ correspond to different weights achieving high value.

Eq. (22) shows that it is trivial to compute $\mathbb{E}[X_0]$ given either (i) the *true score* $\nabla_{x_u} \ln q_u(x_u)$, or (ii) a *trained score model* $s_\phi(x_u,u)$, i.e. a neural network approximation trained via objective (5).
Rows 3 and 4 of Figure 1 show respectively the evolution of $\mathbb{E}[X_0]$ and that of $X_t$ for another trajectory $X_t$, where $X_t$ is simulated and $\mathbb{E}[X_0]$ is computed using the *trained score model* $s_\phi(x_u,u) \approx \nabla_{x_u} \ln q_u(x_u)$ from [1].
Rows 5 and 6 of Figure 1 convey the same information under a reduced number of time-discretization steps: 100 instead of 1000.
All shown trajectories are not handpicked and representative of the population as a whole.

[1] Song, Sohl-Dickstein, Kingma, Kumar, Ermon, Poole -- Score-based Generative Modeling through Stochastic Differential Equations

---

> ### Author Response · Authors · 2021-11-10
> **Addressing Key Issues [part 2]**
>
> ### Addressing Limited Numerical Experiments
>
> Reviewers QAeW and oeyA are concerned with the lack of experimental evidence.
>
> We would as well have preferred to include numerical experiments showing sample quality and reporting quantitative metrics such as FID, but we are currently lacking the computational resources required for training.
> Taking the implementation of [1] as a reference, training on CIFAR10 is carried out over around 3300 epochs, with exponential averaging of weights, following a fine-tuning of the hyperparameters for each proposed model.
> Hence, we focused where it was possible for us to contribute.
> We carried out a number of methodological advances jointly with an analysis of the existing method of [1].
> We only performed numerical experiments requiring limited GPU resources, such as the experiment of Section 5.
> In the meantime we are addressing our computational restrictions with the goal of writing a follow-up work focusing on computational aspects and empirical comparisons.
> We agree that at least a toy experiment is desirable as supporting evidence for the correctness of the proposed method.
> We have thus added a numerical experiment to the (novel) Section 7 of the revised manuscript.
>
> [1] Song, Sohl-Dickstein, Kingma, Kumar, Ermon, Poole -- Score-based Generative Modeling through Stochastic Differential Equations
>
> ### Clarifying the Benefits of Our Proposal
>
> Reviewers QAeW and oeyA find unclear what are the benefits of our proposal, and in what way it would constitute an improvement over alternative approaches.
>
> We agree with the Reviewers that we did not produce any empirical evidence supporting a competitive performance of the proposed method compared to alternative approaches (see previous point).
> It would also be difficult to carry out a practically-relevant theoretical comparative analysis.
> A bound akin to the one established Theorem 1 of [2] would still depend on to-be-estimated constants.
>
> We stress however two main benefits of proposed approach.
> Firstly, in the time-reversal approaches such as [1] the forward process needs to be formulated to obtain an ergodic process (VESDE is not ergodic according the standard mathematical definition, but some arguments can be produced in terms of signal-to-noise ratio).
> As noted in the introduction, our approach offers much greater freedom in the choice of the SDE coefficients $f()$ and $g()$ and of the SDE initial distribution.
> Secondly, while approaches based on the theory of Schrödinger Bridges ([2], [3], [4]) also offer an improved modelling flexibility, the proposed approach is remarkably simpler to train.
> Training is implemented via training objectives of comparable / simpler complexity to the training objective of [1] (Section 6).
> And it is not necessary to implement multiple neural network approximators as in the concurrent ICLR submission [3] or as in [2].
> The novel Section 7 of the revised manuscript details these algorithmic aspects.
> We have also adjusted the Conclusion of the revised manuscript to better convey these advantages.
>
> [1] Song, Sohl-Dickstein, Kingma, Kumar, Ermon, Poole -- Score-based Generative Modeling through Stochastic Differential Equations
>
> [2] De Bortoli, Thornton, Heng, Doucet -- Diffusion Schrödinger Bridge with Applications to Score-Based Generative Modeling
>
> [3] Anonymous Authors -- Likelihood Training of Schrödinger Bridge using Forward-Backward SDEs Theory
>
> [4] Wang, Jiao, Xu, Wang, Yang -- Deep Generative Learning via Schrödinger Bridge

---

> > ### Author Response · Authors · 2021-11-10
> > **Addressing Key Issues [part 3]**
> >
> > ### Addressing Technicality of Presentation
> >
> > Reviewers QAeW and e93q find the presentation overly technical, commenting that the paper is "filled with tons of math", "overburdened by notation" and "written in an obscure way for the non-specialists of this field".
> >
> > Regrettably we were not able to produce a clear presentation for all the reviewers.
> > As this is primarily a methodological paper, some level of formality is inevitable to avoid being grossly imprecise or incorrect.
> > We avoided most of the technicalities by relegating to the Appendices the assumptions, proofs and all the mathematical formulas that did not help the presentation flow.
> > We trade off some precision with ease of exposition by relying on a lightweight notation and a number of conventions detailed in the notation paragraph of the manuscript.
> > To improve clarity, taking into account the Reviewer's remarks, we have revised the manuscript by introducing a clearer notation to disambiguate between the forward and backward time processes in the time-reversal approach (instead of relying on the notation $t$ vs $u$ which is inherently ambiguous) and by expanding the notation paragraph to clarify notation conventions that we took for granted.

---

### Author Response · Authors · 2021-11-18
**Rebuttal Version Uploaded**

We have taken into account the issues raised by all Reviewers and uploaded the revised manuscript.

The major changes are:
- a novel Section 7 which includes: (i) training and sampling algorithm boxes for the proposed DBMT; (ii) a toy numerical experiment to demonstrate the DBMT
- the use of the letter $Y$ to indicate the DTRT process on noising time (thus avoiding ambiguity for the time-direction), various other revisions aimed at improving clarity

As the number of pages available for the rebuttal is still 9, we moved part of Section 8 (revised manuscript numbering) to Appendix C to make space for Section 7.

We are looking forward to engaging in further discussions with the Reviewers to provide additional clarifications if needed.

---

### Decision · Program_Chairs · 2022-01-20

**Decision:**

Reject

**Comment:**

This paper introduces a new method for diffusion-based generative modeling through a Brownian bridge formulation, where the data and latent variable can be coupled. They extend their method to mixtures of diffusion bridges and spatially correlated processes that go beyond the factorial diffusion processes used in prior work.

We thank the authors for engaging with the reviewers and addressing many of their detailed concerns. While reviewers agreed that the proposed theory and methodology were novel and interesting, there are no small or large scale experiments or empirical comparisons to the relevant prior work. In the absence of theoretical justification (bound or proof) as to why the proposed diffusion bridge mixture transport method would result in better performance, more empirical comparisons and evaluations are needed. Additionally, several reviewers found the presentation confusing and overly complex, including the notation, writing, and figures. Given the lack of experimental results and concerns over presentation, I’m inclined to reject this paper.